# Synaptic representation of locomotion in single cerebellar granule cells

**Kate Powell[1][†], Alexandre Mathy[1,2][†], Ian Duguid[1,3]\*, Michael Häusser[1]\***

[1]Wolfson Institute for Biomedical Research and Department of Neuroscience, Physiology and Pharmacology, University College London, London, United Kingdom; [2]Nuffield Department of Clinical Neurosciences, John Radcliffe Hospital, Oxford, United Kingdom; [3]Centre for Integrative Physiology, School of Biomedical Sciences, University of Edinburgh, Edinburgh, United Kingdom

**Abstract** The cerebellum plays a crucial role in the regulation of locomotion, but how movement is represented at the synaptic level is not known. Here, we use in vivo patch-clamp recordings to show that locomotion can be directly read out from mossy fiber synaptic input and spike output in single granule cells. The increase in granule cell spiking during locomotion is enhanced by glutamate spillover currents recruited during movement. Surprisingly, the entire step sequence can be predicted from input EPSCs and output spikes of a single granule cell, suggesting that a robust gait code is present already at the cerebellar input layer and transmitted via the granule cell pathway to downstream Purkinje cells. Thus, synaptic input delivers remarkably rich information to single neurons during locomotion.

*For correspondence:
ian.duguid@ed.ac.uk (ID);
m.hausser@ucl.ac.uk (MH)

[†]These authors contributed equally to this work

**Competing interests:**
See page 16

**Reviewing editor**: Indira M Raman, Northwestern University, United States

## Introduction

In order to ensure generation of precise and reliable movements, information about movement parameters must be represented in neural circuits of the mammalian brain with high fidelity. Firing patterns directly related to specific movement parameters have been reported in single-unit recordings across several brain areas involved in movement representation and generation (*Armstrong, 1988*; *Beloozerova et al., 2003*). Crucial to generating such accurate representations is delivery of synaptic input patterns containing rich information about the animal's movement. However, we currently lack information about the input patterns received during movement for any neuron in the mammalian brain, and thus the input–output transformations performed in these neurons during movement are only poorly understood.

The cerebellum is thought to play a key role in precise limb coordination during voluntary movements (*Flourens, 1824*; *Vercher and Gauthier, 1988*; *Muller and Dichgans, 1994*; *Bastian et al., 1996*; *Holmes, 1922*). Purkinje cells, the output cells of the cerebellar cortex, exhibit firing linked to the phase of the step cycle during locomotion (*Orlovsky, 1972*; *Armstrong and Edgley, 1984*; *Edgley and Lidierth, 1988*). How this information is represented in and transmitted by upstream neurons in the circuit, in particular in cerebellar granule cells—which form the input layer of the cerebellar cortex—remains unknown. Here we have taken advantage of the electrical compactness of granule cells, and their small number of excitatory inputs—4 on average (*Eccles et al., 1967*; *Palkovits et al., 1971*; *Jakab and Hamori, 1988*)—which allows for individual synaptic inputs to be resolved in vivo (*Chadderton et al., 2004*; *Jörntell and Ekerot, 2006*; *Rancz et al., 2007*; *Arenz et al., 2008*; *Chadderton et al., 2014*). Moreover, since granule cell activity plays a key role in regulating locomotion, including the coordination of individual limbs (*Vinueza Veloz et al., 2014*) as well as in in motor learning (*Galliano et al., 2013*), they represent particularly attractive targets for an electrophysiological dissection of their input–output relationships during locomotion.

**eLife digest** Our voluntary movements, such as shaking hands and walking, are controlled by a region of the brain called the cerebellum. Inside this region is a layer of cells called granule cells, which are the smallest and also the most numerous type of neuron in the brains of mammals. Granule cells receive information from many other parts of the brain and respond by producing electrical signals that influence the motor system, which tells our muscles how to move. However, it is not clear how the granule cells interpret the information they receive and ensure that the right muscles are stimulated at the right time by the motor system.

Powell et al. have now used 'patch-clamp electrodes' to measure the electrical activity of individual granule cells in the cerebellum of mice, both at rest and as they walked. This is a powerful approach as it enables the recording of both the information received by each granule cell (input) and the electrical signals produced by it in response (output). Each mouse was placed on a treadmill with its head held still and given the choice to either rest or walk. These experiments show that when the mouse is resting, the granule cells are mostly inactive, producing only very low levels of fast electrical signals called 'spikes'. When the mouse starts walking, the input to the granule cells triggers a strong increase in spiking in the granule cells.

Powell et al. used a computer model to understand how the granule cells represent movement. Remarkably, this model could be used to predict walking patterns of the mouse based on the activity of a single granule cell and its inputs. These findings suggest that even single neurons in the cerebellum contain rich information about the movement of the animal. The next challenge is to understand how this code interacts with the rest of the motor system to produce precisely coordinated movements. Furthermore, it will be important to determine whether a similar code is used in other parts of the brain that control movement.

By recording the activity of mossy fiber boutons, EPSCs in granule cells, and granule cell output while mice are moving on a treadmill, we can thus reconstruct single cell integration of synaptic inputs in awake animals during locomotion, and identify the cellular representation of movement parameters in a defined site in the circuit.

## Results

### Whole-cell recordings from granule cells and mossy fibers during locomotion

In vivo whole-cell recordings were made from mossy fiber boutons and cerebellar granule cells in lobule V of the cerebellar vermis. All recordings were performed in awake mice head-fixed on a spherical treadmill (*Figure 1A*). Granule cells and mossy fiber boutons were identified on the basis of their distinctive electrophysiological signatures (*Chadderton et al., 2004*; *Jörntell and Ekerot, 2006*; *Rancz et al., 2007*; *Arenz et al., 2008*). To study the link between voluntary movement and granule cell input and output we extracted a motion index from captured video frames (*Figure 1B*, see 'Materials and methods') and aligned this to the simultaneously acquired electrophysiological data (example recordings *Figure 1C–E*). The motion index was used to categorize the electrophysiological data recorded during quiet wakefulness (defined as periods where the motion index remained below a threshold rate of change of 0.025 a.u. per frame, for at least 30 consecutive frames, see 'Materials and methods') and voluntary movement.

During quiet wakefulness, granule cells exhibited a resting membrane potential of $-67.0 \pm 8.9$ mV and an input resistance of $420 \pm 210$ MΩ (n = 47, in 24 mice). Recordings from mossy fiber boutons (*Rancz et al., 2007*) demonstrated a spontaneous average firing frequency of $21 \pm 20$ Hz (n = 6, in 5 mice). The average frequency of spontaneous excitatory synaptic currents in granule cells was $60 \pm 35$ Hz (n = 32, in 16 mice). Despite this relatively high input frequency, granule cell output spiking frequency was low: the average firing rate was $0.12 \pm 0.31$ Hz (n = 27, in 13 mice), and 62% of granule cells showed no spiking during periods in which the mouse was sitting quietly.

In some cases we were able to successfully maintain recordings of activity during periods of voluntary locomotion (ranging from 17 s to 65 s, mean duration: $33.2 \pm 12.3$ s). This allowed to us to

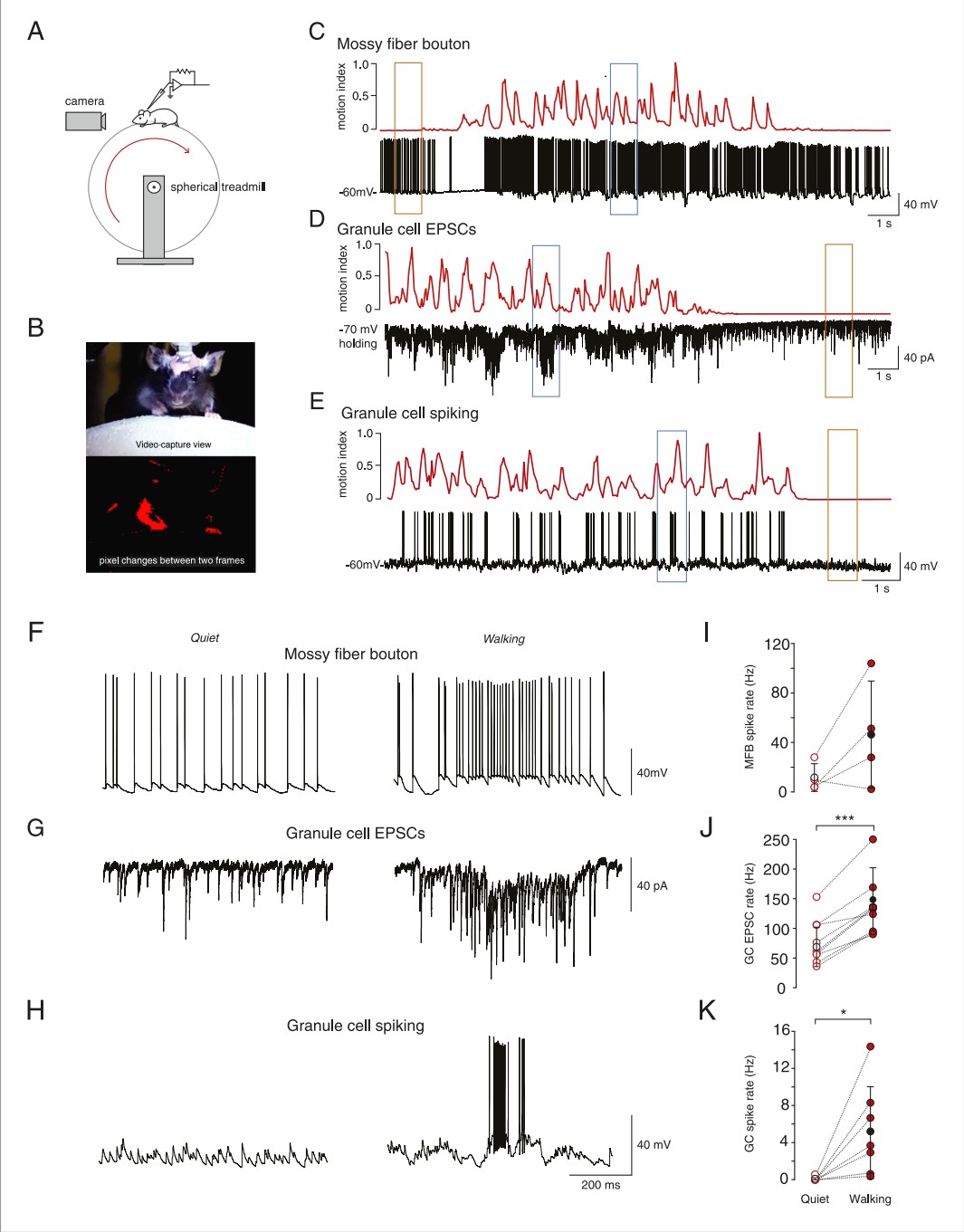

**Figure 1**. Whole-cell recordings from granule cells and mossy fibers during locomotion. (**A**) Schematic of recording configuration. (**B**) Calculation of motion index. Top panel: a single frame from a video of a mouse walking on the treadmill. Bottom panel: pixel intensity variation between the frame shown in the above panel and the previous frame are highlighted in red. The pixel variation between each frame was quantified for each video to give a continuous signal relating to average motion of the mouse (motion index calculated as described in the 'Materials and methods' and normalized to the maximum value in the video). (**C–E**) Example whole-cell recordings (black) from a presynaptic mossy fiber terminal (**C**), a granule cell recorded in voltage-clamp mode (**D**) and a granule cell recorded in current clamp mode (**E**), together with the corresponding motion index (red). (**F–H**) Section of each example trace shown in **C–E** at a higher timescale show spontaneous input recorded during a quiet period (left panels, orange frames in (**C–E**) indicate the location within the trace) and typical bursts of activity during locomotion (right panels, blue frames in (**C–E**) indicate the location within the trace). Summary data comparing the average instantaneous frequencies of mossy fiber spikes (**I**, n = 4 in 4 mice), granule cell EPSCs (**J**, n = 9 in 6 mice)

*Figure 1. continued on next page*

*Figure 1. Continued*
and granule cell spikes (**K**, n = 6, in 4 mice). Mean group averages are indicated with black open and closed circles, error bars indicate standard deviation.

compare activity within single cells during quiet periods and during locomotion. Locomotion was accompanied by an increase in mossy fiber input to the granule cells, observed as an increase in firing frequency in 3 out of 4 recorded mossy fibers, although overall this was not statistically significant across the four recordings (*Figure 1F,I*; average, quiet: 11.7 ± 11.6 Hz; locomotion: 46.4 ± 43.3 Hz, n = 4 in 4 mice, p = 0.14, paired t-test) and increased EPSC frequency in all recorded granule cells (*Figure 1G,J*; quiet: 77.5 ± 37.6 Hz; locomotion: 144.0 ± 53.5 Hz, n = 9 in 6 mice, p < 0.0001, paired t-test). The firing rate of the granule cells also increased dramatically during locomotion (*Figure 1H,K*, average; quiet: 0.09 ± 0.2 Hz; locomotion: 5.3 ± 5 Hz, n = 7 in 4 mice, p < 0.05, paired t-test). During locomotion, granule cell action potentials occurred in sparse high-frequency bursts with high instantaneous firing frequencies (*Figure 1H* right panel, average instantaneous frequency: 106 ± 65 Hz; 4 out of 7 cells fired in bursts, defined as groupings of 4 or more spikes with ISIs less than 50 ms). These bursts consisted of an average of 11.9 ± 2.1 spikes with ISI 10.7 ± 3 ms, occurred with an average inter-burst interval of 1.88 ± 0.99 s, and were associated with a high coefficient of variation of the inter-spike interval (CV = 2.2 ± 1.1). The CV of the inter-spike interval was significantly higher than the CV of the EPSC inter-event interval during movement (1.1 ± 0.3, n = 9, p = 0.039, unpaired t-test). We used the bootstrap method (*Roy, 1993*) to calculate 95% confidence intervals on the GC EPSC/MFT spike ratio, which were 3.5–17.7 at rest, and 1.7 to 8.4 during motion, consistent with the known 4:1 anatomical convergence (*Eccles et al., 1967*; *Palkovits et al., 1971*; *Jakab and Hamori, 1988*).

## Glutamate spillover currents drive spiking during locomotion

The sustained high-frequency barrages of EPSCs during locomotion (*Figure 1G*, right panel) were associated with large, slow inward currents, comparable to the slow spillover currents that have been observed at this synapse in vitro (*DiGregorio et al., 2002*; *Xu-Friedman and Regehr, 2003*; *Nielsen et al., 2004*). NMDA receptor activation could in theory underlie such slow currents; however the NMDA current is negligible in cerebellar granule cells at our −70 mV holding potential, due to the voltage-dependent $Mg^{2+}$ block of the NMDA receptor channel (*Nieus et al., 2006*; *Figure 2*). We confirmed that the NMDA current is negligible using a GC model (*Diwakar et al., 2009*), adapted to match the NMDA/AMPA ratio of 0.2 found in adult mice (*Cathala et al., 2003*). At −70 mV, NMDA contributes at most 0.4% of the total excitatory synaptic current in this model. Furthermore, it has previously been shown (*Cathala et al., 2003*) that in the age and species of mice from which our data is derived, the vast majority of the NMDA current is transmitted by extrasynaptic receptors. We therefore concluded that spillover transmission must be responsible for most of the slow current we measured.

To isolate the putative spillover components, we fit the fast EPSCs and separated them from the underlying slow current recorded at −70 mV (*Figure 2A*). The amplitude of the slow current was correlated with the EPSC frequency (*Figure 2E* shows this for an individual cell; *Figure 2F* for the population of n = 9 cells), such that the proportion of the total current carried by the fast EPSCs vs the slow current diminished with EPSC rate (*Figure 2B*), similar to what has been shown in vitro in response to trains of synaptic stimulation (*Sargent et al., 2005*). Since spillover results from synaptic accumulation of neurotransmitter, a delay would be expected between the peak of the EPSCs and that of the putative spillover current. Indeed, a cross-correlation between the instantaneous frequency of fast EPSCs and the slow currents showed a peak with a lag of 40 ms (*Figure 2C*), with the slow current lagging the fast events. Similarly, aligning the slow current on a burst of at least 5 EPSCs at 200 Hz or above exhibited a negative peak at positive delay (31 ms latency; 729 bursts from the 9 cells, *Figure 2D*). We further plotted the synaptic charge transfer over 100 ms as a function of EPSC rate with and without spillover (*Figure 2—figure supplement 1*). The spillover dramatically affects the slope of this curve.

What is the functional role of the slow putative spillover current? In current clamp recordings from granule cells, we observed that spiking occurs in bursts (*Figure 2G*; 37 bursts recorded in 4 out of 7 cells), which were frequently accompanied by a slow depolarization (*Figure 2G*, left panel). As predicted, there was a significant correlation (r = 0.45, p = 5.5 × 10⁻³; *Figure 2H*) between the

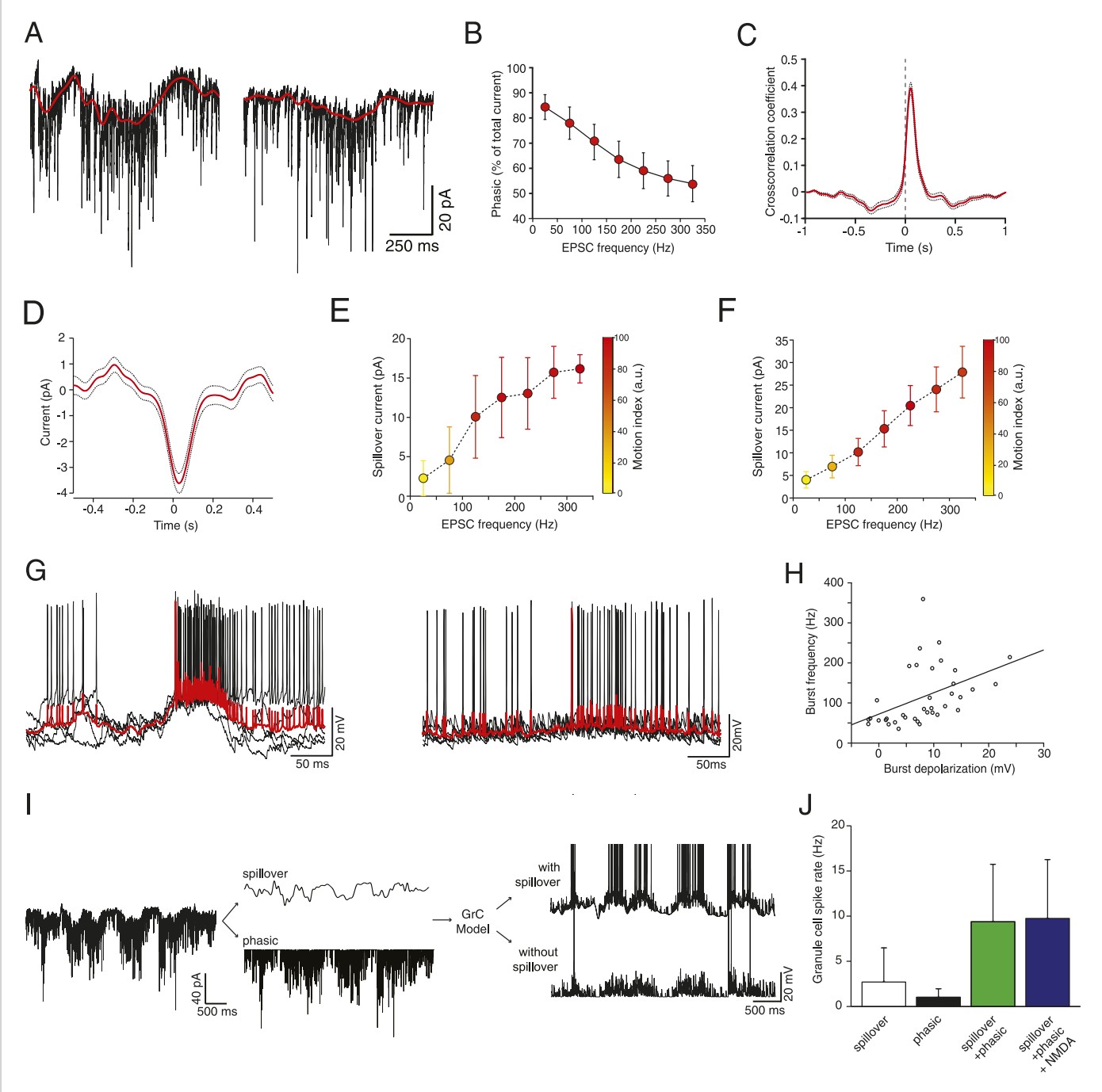

**Figure 2**. Glutamate spillover enhances transmission during locomotion. (**A**) Voltage clamp traces recorded at a holding potential of −70 mV from two granule cells. Red line indicates the spillover current obtained by subtracting the fitted single exponential decays of the phasic EPSCs. (**B**) Relationship between the EPSC frequency and the relative proportion of excitatory current carried by the phasic EPSC component (average for all recorded cells; n = 9 in 6 mice). (**C**) Average cross-correlation of instantaneous EPSC frequency with spillover current (n = 9, dotted lines indicate standard deviation). Dashed line indicates zero lag. (**D**) Average EPSC burst triggered spillover current for all cells (dotted line indicates standard deviation). (**E**, **F**) Graphs showing increasing EPSC frequency and spillover conductance occurring with motion in an example granule cell (**E**) and as an average for all cells (**F**, n = 9). (**G**) Overlaid voltage traces showing spike bursts (red traces indicate the average). High-frequency bursting appears to be associated with greater subthreshold depolarization (left panel). (**H**) Graph showing the relationship between spike burst frequency and subthreshold depolarization for individual bursts across all cells (n = 37 bursts from 4 cells, r = 0.45, p = 0.0055). (**I**) A representative current trace showing the separation of phasic and spillover EPSCs. A granule cell model was then used to estimate the effect of spillover conductance on granule cell spike output. (**J**) Summary data showing the
*Figure 2. continued on next page*

*Figure 2. Continued*
simulated granule cell spike frequency resulting from spillover, phasic and combined conductances, as well as combined conductances including an NMDA conductance (n = 9; see 'Materials and methods').
The following figure supplement is available for figure 2:

**Figure supplement 1**. Synaptic charge transfer with and without spillover.

amplitude of the slow depolarization and the spike frequency in the burst. We hypothesized that the spillover could contribute to such spiking episodes. To study this, we used a spiking compartmental model of a granule cell (*Diwakar et al., 2009*), with an excitatory synaptic conductance based on the fast EPSCs and slow spillover currents recorded in voltage clamp. The depolarization generated by the slow spillover current was able to facilitate bursts that were not observed with fast EPSCs alone (*Figure 2I*). Consistent with this, the combination of spillover and fast currents produced significantly more spikes than either the spillover or fast current in isolation, indicating a synergistic non-linear interaction between the two currents (spiking rate for spillover alone: $2.69 \pm 3.78$ Hz, for fast EPSCs: $1.01 \pm 0.94$ Hz, for both: $9.27 \pm 6.36$ Hz, *Figure 2J*, $p = 3.39 \times 10^{-6}$; one way ANOVA). Next, we assessed the possible contribution of NMDA receptor currents to spiking under these conditions. We performed simulations using an NMDA to AMPA ratio of 0.2, which is physiological at this synapse (*Cathala et al., 2003*). Under these conditions a spiking rate of $9.74 \pm 6.51$ Hz is obtained, which is only slightly higher than without the NMDA channels (rate $9.37 \pm 6.35$ Hz, $p = 2.70 \times 10^{-8}$; paired t-test). We conclude that the slow putative spillover current contributes significantly to spike generation in awake, locomoting animals.

## Linking synaptic input parameters with output spiking during locomotion

We next wanted to study the relationship between activity in the granule cell layer and movement parameters. The mice were free to initiate locomotion at will, and were mostly quietly resting on the treadmill. We examined how EPSC activity changes at the start of these periods of locomotion, and found a sustained increase in rate from $100 \pm 8$ Hz to $145 \pm 12$ Hz ($p = 0.0036$, paired t-test, n = 23 periods from 9 cells comparing a 500 ms window 1 s before and immediately after locomotion onset; *Figure 3—figure supplement 1*). Similarly, at the termination of locomotion, there was a sustained decrease in EPSC rate from $139 \pm 10$ Hz to $106 \pm 7$ Hz ($p = 0.0118$, paired t-test, n = 23 periods from 9 cells comparing a 500 ms window immediately before and 1 s after locomotion onset). Note however that even in the absence of locomotion, EPSCs occurred at high frequencies.

Next, we examined the precise relationship between activity in mossy fibers and granule cells during locomotion. Positive correlations were found between the motion index and MFB spiking, GC spiking, GC EPSC rate and spillover current in most cells at low temporal resolution (1.5 s; *Figure 3C*). At higher temporal resolution significant peaks in the normalized sliding cross-correlations were observed between motion and mossy fiber bouton spiking, granule cell EPSCs and granule cell spiking (*Figure 3A,B*). For spillover, only a minority of the cells (2 out of 9) showed a significant peak cross-correlation at high temporal resolution, consistent with it being a slow current. Together, these results indicate that overall motion during locomotion is represented in both granule cell excitatory input and spike output patterns.

To correlate specific movements with granule cell input and output, we performed activity-triggered averaging of motion maps in video images of locomoting mice (*Figure 3D*). Anatomically coherent regions of the mouse showed strong signals (19 out of 20 of these maps showed significant signals in the regions corresponding to one or more of the limbs, 9 out of 20 for the head, and 11 out of 20 for the body of the mouse), suggesting that more specific features of the motion could be read out by granule cell activity. We therefore decomposed the videos into principal components (PCs), which corresponded to motions of the mice, most often showing limb motion and/or locomotion at different speeds (*Figure 3E*). For all the 20 cells recorded, we used a semi-automated labelling algorithm (see 'Materials and methods') to classify the 50 highest eigenvalues and found that 738 out of 1000 showed periodic limb motion, 434 showed periodic head motion, and 370 showed periodic motion of the rest of the body. We performed a sliding cross-correlation of the electrophysiological

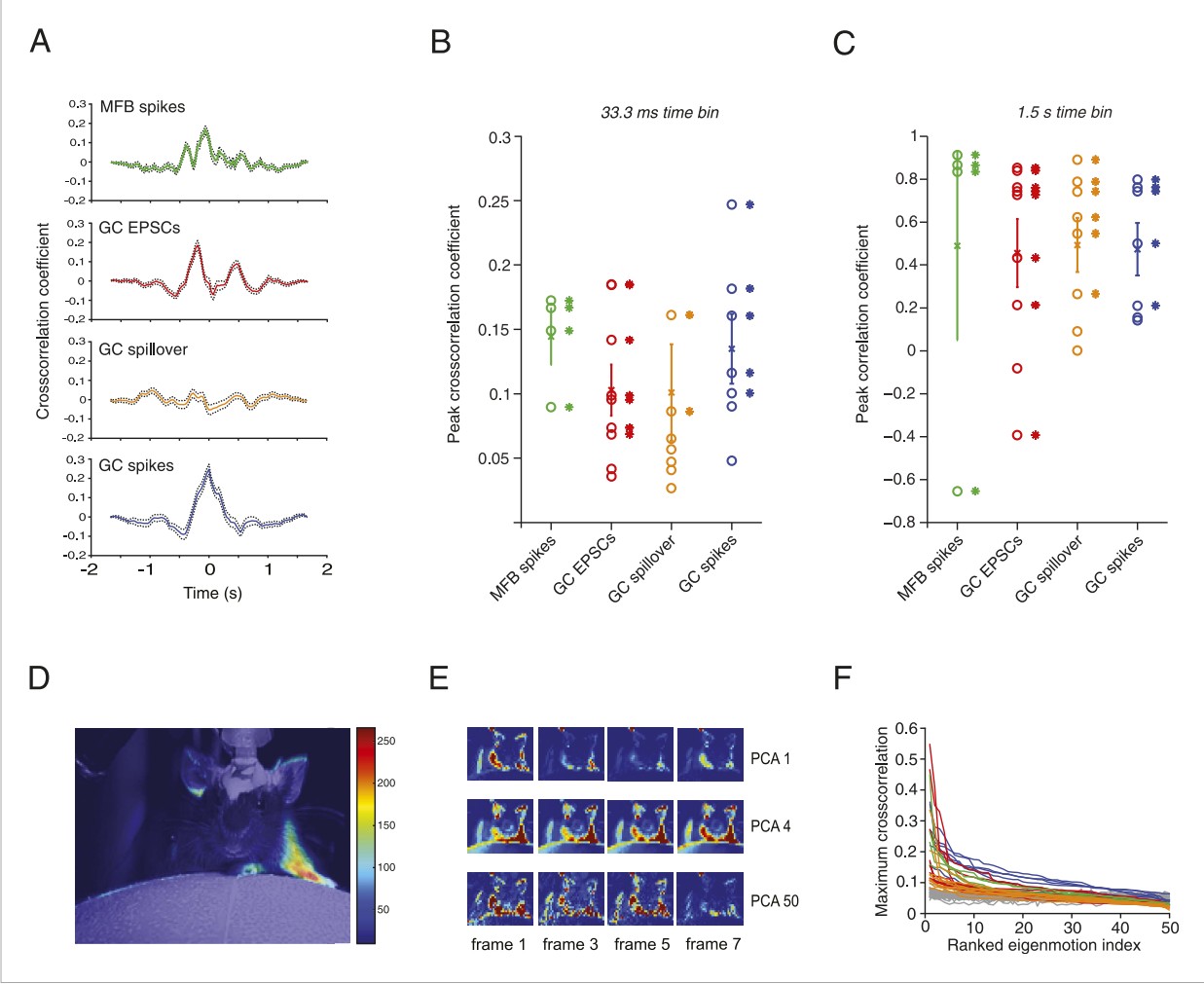

**Figure 3**. Relationship between motion and activity parameters of single granule cells. (**A**) Normalized cross-correlation between motion index and event frequency for a MFB (green), EPSCs (red), spillover (orange), and GC spikes (blue). (**B**, **C**) Peak normalized cross-correlation coefficients for each cell using fine time bins (33.3 ms; asterisks denote points with p < 0.05) and raw correlation coefficients using coarse time bins (1.5 s; asterisks in **C** denote correlation coefficients significant at the p = 0.05 level). (**D**) Spike triggered average of motion map overlaid on video of mouse. (**E**, **F**) PCA analysis: each video was decomposed into 50 8-frame principal components (PCs, top row, 4 frames shown for PC 1, 4 and 50). The activity of each cell was then cross-correlated over time with the PC coefficient (**F**). Each line represents the maximum cross-correlation of a single cell with the different PCs for the corresponding video (green = MFBs, red = GC-EPSC, orange = GC-spillover, blue = GC-spikes). The gray traces represent the average cross-correlation with bootstrapped data and therefore indicate the noise level. Note that the PCs were ranked according to their cross-correlation.

The following figure supplement is available for figure 3:

**Figure supplement 1**. Modulation of responses by locomotion onset and termination.

activity with the projection of the video onto the 50 PCs with the largest eigenvalues. For all recording modalities, peak cross-correlation higher than 0.3 and up to 0.55 could be found (*Figure 3F*); these correlations dropped below 0.2 after the 10th component. Note that the correlations were higher than with surrogate data (grey traces, *Figure 3F*), which indicates that the neural activity correlated with specific PCA components more than would be expected by chance.

## A simple model based on single granule cell activity predicts the step cycle

Next, we examined the relationship of activity parameters to the step cycle of the mouse, focusing on the two forelimbs. Animals initiated periods of locomotion spontaneously, and would often be still for several

seconds between these periods. Electrophysiologically recorded parameters in mossy fiber boutons and granule cells were strikingly modulated by the step cycle (*Figure 4A–C*), in either or both of the forelimbs. Specifically, statistically significant step cycle modulations were observed for mossy fiber bouton spiking (3 out of 4 cells for the right limb, 1 out of 4 cells for the left limb), granule cell EPSCs (5 out of 9 for the right limb, 4 out of 9 cells for the left limb), and granule cell spiking (0 out of 7 for the right limb, 2 out of 7 cells for the left limb); while spillover was not significantly correlated with step cycle. In *Figure 4—figure supplement 1*, we show why it is possible that despite high cross-correlations between spillover and EPSCs, significant step cycle modulations are not seen: a temporally filtered EPSC

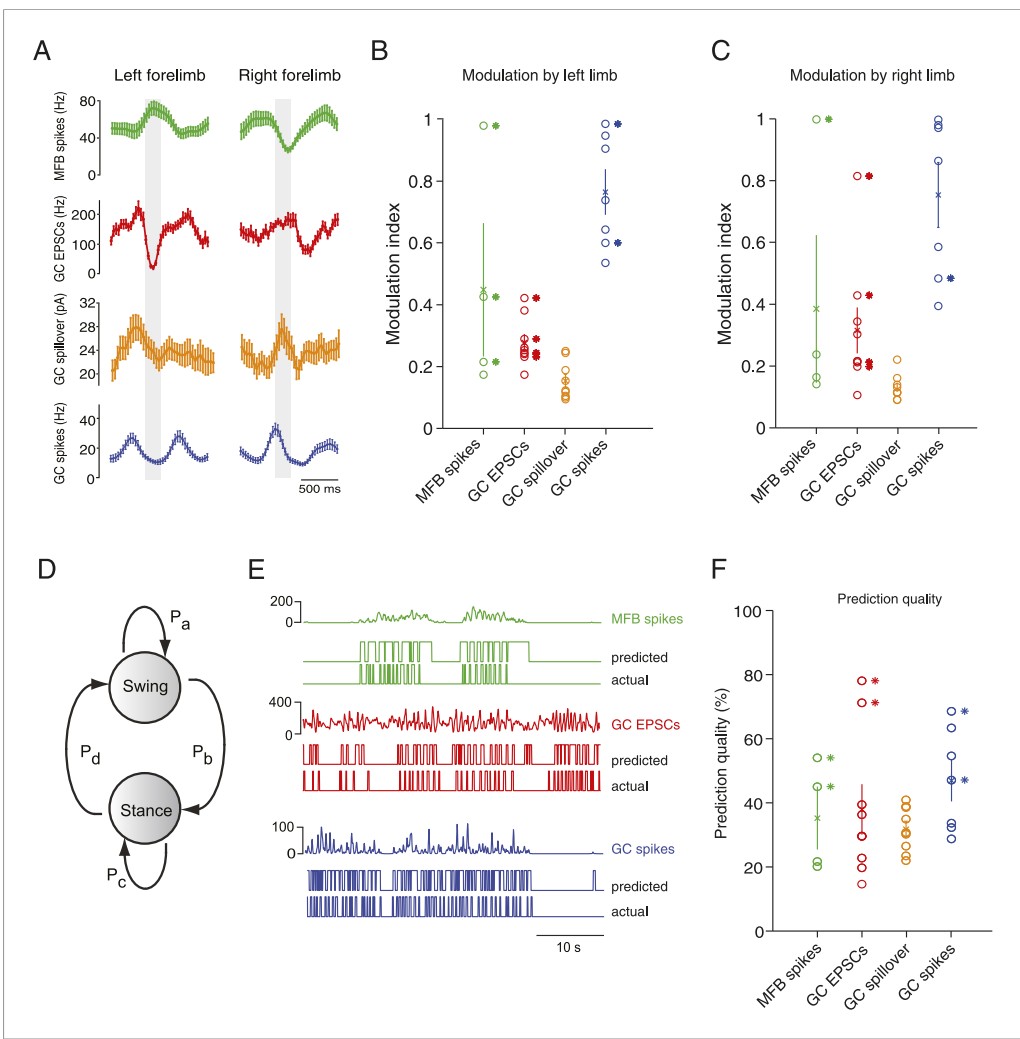

**Figure 4**. Decoding activity in a single granule cell can predict the step cycle. (**A**) Example step-triggered averages of activity for three different cells (EPSC and spillover examples are from the same granule cell). Gray-shaded area indicates the swing phase of the step cycle. (**B**, **C**) Step cycle modulation index for each forelimb across all cells. (**D**) Two state Hidden Markov Model (HMM) used to reconstruct the step cycle from electrophysiological recordings. (**E**) Example of successful step reconstructions for an MFB (red), GC EPSC recording (green) and GC spikes (blue). The top traces represent the electrophysiological event rate in Hz, the middle traces the step transition (with the high state being the swing phase and the low state being the stance) predicted by the HMM, and the lower trace being the actual step of the best modulated limb. (**F**) Prediction quality for all cells.

The following figure supplements are available for figure 4:

**Figure supplement 1**. Spillover acts like a temporal filter.

**Figure supplement 2**. Tuning of responses to the step cycle.

trace (using a biexponential kernel with rise time = 50 ms and decay = 100 ms) exhibits a cross-correlation with the original trace which is similar to spillover (*Figure 4—figure supplement 1B*), however its step cycle modulations are significantly lower (from $0.30 \pm 0.03$ to $0.13 \pm 0.2$, $p = 4.65 \times 10^{-4}$, paired t-test; n = 18—that is, 2 limbs for n = 9 cells), due to the low-pass properties of the filter.

To examine in greater detail the tuning of events to the step cycle, we plotted the step cycle modulation of the event rates on a polar plot for both forelimbs. Mapping the start of the step cycle to 0˚ of the circle, we observed that cells could show maximal modulation in a wide range of phases (*Figure 4—figure supplement 2*; range from 0 to 351˚). The phase difference between the maximal modulation for one forelimb vs the other was clustered around 90˚: $102 \pm 30$˚ for MFTs, $104 \pm 19.2$ for EPSCs, $74 \pm 15$ for GC spikes. Not surprisingly, the magnitude of modulation was highly correlated between the two forelimbs (n = 20 cells, R = 0.9, $p = 3.48 \times 10^{-8}$). However as the MFT example in *Figure 4A* makes clear, not all modulations of activity were in the same direction for both forelimbs. In the right panel of *Figure 4—figure supplement 2C*, we have plotted the maximum modulations as z-scores for one limb vs the other, which shows across the population that modulations can happen in all four directions (up-up, up-down, down-up, down-down).

If these step-related modulations are sufficiently robust to provide a code for downstream Purkinje cells, it should be possible to reconstruct the step sequence from the recorded activity. We therefore implemented a two-state Hidden Markov Model (HMM; *Rabiner, 1989*), with one state representing the stance and the other state representing the swing phase of the step cycle (*Figure 4D*), and trained this model on the electrophysiological data. Note that the algorithm used to constrain the model is unsupervised in the sense that the learning procedure is never shown the actual step sequence. A predicted step sequence was generated for each trace by producing the most likely state sequence for the HMM trained on the trace. This prediction was then compared to the actual step sequence. For mossy fiber spikes, granule cell spikes, and granule cell EPSCs we found that in some cells (2 out of 4, 2 out of 5 and 2 out of 9 cells respectively, *Figure 4E*) accurate step sequence reconstructions (i.e., with p-values < 0.05) could be obtained (*Figure 4F*). In contrast, step sequences could not be reconstructed from spillover currents alone, indicating that while these currents are permissive for spiking during movement, they do not represent the details of individual movements. Together, these findings indicate that rich information about locomotion parameters is available to single granule cells in the cerebellar cortex, even at the level of synaptic input.

## Discussion

We have directly measured synaptic input patterns and spike output in single neurons in awake mice, both at rest and during locomotion. Our experiments reveal that granule cells exhibit low firing rates in awake animals at rest, but high-frequency spiking during locomotion. Mossy fiber boutons can also exhibit dramatic increases in spiking driven by locomotion, which drives high-frequency bursts of EPSCs in granule cells, associated with a large glutamate spillover component that enhances spike output during locomotion. Surprisingly, locomotion can be highly correlated with the activity of single mossy fibers, EPSCs in single granule cells, and output from single granule cells. We demonstrate that a HMM allows the step sequence to be reconstructed from the activity of only a single mossy fiber; or EPSC input; or the spike output of a single granule cell. These findings provide crucial new insights into the synaptic representation of movement parameters, and how the cerebellar cortex processes movement.

### Activity of granule cells and their synaptic inputs in awake animals at rest and during locomotion

Our experiments represent the first whole-cell recordings from granule cells in awake mice. Information about granule cell activity patterns in awake animals has been extremely difficult to obtain. Extracellular recordings from the granule cell layer have suffered from the fact that the exceptionally dense packing of the granule cells has made it difficult to unambiguously assign spikes as arising from single granule cells (*Hartmann and Bower, 2001*; *Gao et al., 2012*). Although two-photon imaging approaches should in principle help to resolve this issue, the scattering nature of the densely packed granule cell layer, and the lack of reliable single-spike sensitivity of current genetically encoded calcium sensors combined with the difficulty of separating synaptic and action potential-linked calcium signals in granule cells has made it difficult to reliably estimate granule cell firing rates using two-photon imaging (*Ozden et al., 2012*).

Our patch-clamp recordings allowed us to unambiguously record from single granule cells, and demonstrate that granule cells in awake mice exhibit remarkably low firing rates when the animal is at rest, similar to those observed in mormyrid fish under paralysis using similar techniques (*Sawtell, 2010*). Moreover, the electrical compactness of the granule cells allows us to make voltage clamp recordings to probe synaptic currents driving spiking activity. We demonstrate that the low spontaneous firing rates observed at rest in granule cells, which are surprisingly similar to those in anaesthetized (*Chadderton et al., 2004*; *Duguid et al., 2012*) and decerebrate (*Jörntell and Ekerot, 2006*) preparations, persist despite far higher spontaneous EPSC rates than are present under anaesthesia. This indicates that the low firing rate of granule cells is a general property of granule cells and suggests that it is under tight control by intrinsic mechanisms and by synaptic inhibition (*Duguid et al., 2012*), especially in the awake animal.

Despite the low firing rates observed at rest, granule cells exhibited a dramatic increase in firing at the onset of locomotion, paralleling observations using two-photon calcium imaging in the granule cell layer (*Ozden et al., 2012*). This increase in granule cell firing rate was widespread across the granule cell population, indicating that granule cells switch from a sparse to a dense mode of activation during locomotion. The high firing rates of granule cells during locomotion was organized in bursts, reminiscent of the high-frequency bursts observed in granule cells with some forms of sensory stimulation (*van Beugen et al., 2013*), which in turn can be transmitted reliably to the Purkinje cell (*Valera et al., 2012*).

Our voltage clamp experiments reveal that the locomotion-evoked spiking is driven by a significant increase in excitatory synaptic input to the granule cells, which is paralleled by an increase in spiking in mossy fibers, as shown by direct recordings from mossy fiber boutons. Some of the increase in mossy fiber input may be provided by activity in spinocerebellar pathways, which exhibit locomotion-related activity (*Arshavsky et al., 1972a*, *1972b*; *Orsal et al., 1988*). These results indicate that the sensorimotor computations in neuronal circuits upstream from the mossy fiber pathways provide sufficiently strong activation to overcome the high threshold for output spiking in the granule cells. Future experiments are required to examine how the interaction between mossy fiber input and Golgi cell inhibition (*D'Angelo and De Zeeuw, 2009*) jointly determine the enhanced spiking output of granule cells during locomotion.

## The importance of glutamate spillover during locomotion

Our experiments provide the first investigation of glutamate spillover in the awake behaving animal, and demonstrate that this feature of mossy fiber-granule cell transmission (*DiGregorio et al., 2002*; *Nielsen et al., 2004*) plays a crucial role in driving movement-related output spikes from the granule cell layer. Previous investigation of glutamate spillover currents has been restricted to brain slices, both in hippocampus (*Kullmann and Asztely, 1998*) and cerebellum (*DiGregorio et al., 2002*; *Nielsen et al., 2004*), and it has been unclear whether the stimulus conditions and levels of glutamate uptake present in vitro are representative of the physiological pattern of in vivo activation. We demonstrate that spillover currents make a substantial contribution to the synaptic charge during locomotion-evoked mossy fiber input. Furthermore, using a realistic compartmental model of the granule cell, we show that these spillover currents are essential for driving the observed output firing rates of the granule cell associated with locomotion.

The biophysical properties of spillover currents appear to be ideally suited to help ensure appropriate granule cell firing patterns under resting conditions and during locomotion. Specifically, spillover currents are synergistically enhanced by the activation of multiple neighbouring synapses (*Carter and Regehr, 2000*; *Arnth-Jensen et al., 2002*; *DiGregorio et al., 2002*), as is likely during the high-frequency barrage of mossy fiber inputs activated during locomotion. Thus, the requirement of the spillover current for high spiking rates provides another mechanism to both ensure sparse granule cell firing at rest, and enhance firing during locomotion, increasing the signal-to-noise ratio of granule cell transmission of locomotion information.

## Functional implications

We show that granule cells exhibit very low spike rates in awake resting mice, despite high excitatory synaptic drive. Since granule cells represent the most abundant neuronal type in the brain, and spikes are energetically expensive (*Attwell and Laughlin, 2001*; *Carter and Bean, 2009*), the metabolic cost

of sustained firing in this population would be considerable (*Howarth et al., 2012*). Therefore the ability to maintain sparse firing in granule cells at rest could be an important mechanism for energy conservation in the mammalian brain. During locomotion, however, spike rates can increase dramatically. We demonstrate that movement during locomotion exhibits an unexpectedly strong representation in mossy fiber input received by granule cells, as well as in the spiking output of individual granule cells. Remarkably, in some neurons it is possible to reconstruct the entire step sequence during locomotion from both input patterns and spike output, indicating that gait information is present even at the level of single granule cells. This indicates that a rich amount of information about movement parameters is already represented by individual synapses at the input layer of the cerebellar cortex. Thus, selective sampling of granule cells exhibiting tuning to a given phase of the step cycle may underlie the step cycle modulation of simple spikes observed in Purkinje cells (*Orlovsky, 1972*; *Armstrong and Edgley, 1984*; *Edgley and Lidierth, 1988*). Our results suggest that the synaptic representation of locomotion, rather than being based on a sparse code (*Marr, 1969*; *Attwell and Laughlin, 2001*; *Howarth et al., 2012*), relies on a population of tuned neurons, which shift from sparse activity to a dense gait code during movement.

## Materials and methods

### Surgical procedures

All experiments were carried out in accordance with UK Home Office regulations. Adult C57BL/6J mice (P40–60) were anesthetized with isoflurane (3–5% for induction, 0.5–2% for surgery, in pure oxygen). Throughout general anaesthesia, rectal temperature was monitored and body temperature maintained constant using a homeothermic blanket (FHC). Mice were placed in a stereotaxic frame, which allowed horizontal positioning of the head and implanted with a lightweight L-shaped metal head plate and recording chamber. The recording chamber was positioned directly above lobule V of the cerebellum. This region of the cerebellum was the most accessible, requiring the least amount of muscle removal, which was an important consideration for these awake experiments. Mice were allowed to recover from head implant surgery for a minimum of 48 hr.

At least 3 hr prior to performing electrophysiological recordings, mice were re-anaesthetized with isoflurane (as described above). A small (~200–500 µm) craniotomy was created through the skull directly above lobule V of the cerebellar vermis roughly 0.5–1 mm lateral of the midline and the dura removed. Agar and a silicone elastomer (Kwik-Cast, World Precision Instruments Ltd, Sarasota, FL) were applied to the craniotomy to seal it. When the silicone was fully set the mouse was removed from the head-holder and placed in a warm cage to recover from anaesthesia for ~2 hr.

Following recovery from anaesthesia mice were head-fixed ~1 hr prior to starting the recording session. Mice were placed on a 20 cm diameter polystyrene ball that was secured to an air-table directly below and slightly in front of the headstage. The ball was mounted through its center on a horizontal axle resting on bearings. Mice were placed on the ball such that they could rest comfortably on its center and walk voluntarily. With this configuration mice habituated readily to head restraint, usually sitting quietly after 30–60 min.

### In vivo patch-clamp recording

In vivo whole-cell voltage-clamp and current-clamp recordings were obtained from cerebellar granule cells and mossy fiber boutons as previously described (*Chadderton et al., 2004*; *Rancz et al., 2007*; *Duguid et al., 2012*). Recordings used for analysis lasted from 100 s to 950 s (mean average: 300 ± 186 s, n = 53, in 26 mice). Pipettes had resistances of 6–7 MΩ and were filled with an internal solution containing (in mM): K-Methanesulphonate, 133; KCl, 7; HEPES, 10; Mg-ATP, 2; Na$_2$-ATP, 2; Na$_2$-GTP, 0.5. EGTA, 0.1; pH 7.2.

Granule cells were identified based on their characteristically small capacitance and depth from the pial surface (450–600 µm). Mossy fiber boutons were identified by their relatively high spontaneous spike rates, lack of synaptic input and characteristic spike waveform (*Rancz et al., 2007*). Electrophysiological measurements were amplified using a Multiclamp 700B amplifier (Axon Instruments, Molecular Devices, Sunnyvale, CA). Data was filtered at 4–10 kHz and acquired at 20 kHz using pCLAMP software in conjunction with a Digidata 1440A acquisition system (Axon Instruments, Molecular Devices).

## Electrophysiological data analysis

Electrophysiological data was inspected for artifacts relating to movement (large perturbations either side of the baseline). Recordings that demonstrated such artifacts were excluded from further analysis. Electrophysiological data were analyzed using custom-written macros in Igor Pro 6. Synaptic currents and potentials were detected using an amplitude threshold algorithm where the threshold for event detection was set at two times the standard deviation of the baseline noise (typically about 10 pA). Detected currents and potentials were verified manually through careful inspection of all electrophysiological data.

## Video analysis

Videos were acquired using a Canon Exilim-F1 digital camera at a frame rate of 30 Hz. The electrophysiological traces were synchronized with the video by aligning the recordings with the onset of an LED timed by the electrophysiology acquisition software. Analysis of the video was performed using custom built software in Matlab (MathWorks) which is available as *Source code 1*. The video was first cropped to contain a small area displaying the mouse and then used to calculate a motion map as follows: First a background (corresponding to pixels that have not moved recently) was calculated by the following formula:

$$BG_i[x, y] = \alpha \times Frame_i[x, y] + (1 - \alpha) \times BG_{i-1}[x, y],$$

where $BG_i$ is the background at time point i, and $Frame_i$ is the video frame. Empirically, we found $\alpha = 0.3$ to perform well with our dataset, this causes the background to be a weighted average over about the last 15 frames.

Next, a difference image was calculated as follows:

$$diffim_i[x, y] = |Frame_i[x, y] - BG_i[x, y]|.$$

To provide a less noisy estimate of motion, we used a time smoothed difference image as the motion map:

$$motion\_map_i[x, y] = \beta \times diffim_i[x, y] + (1 - \beta) \times motion\_map_{i-1}[x, y],$$

where $\beta = 0.9$.

Spike-triggered averages of this motion map were computed by binning the neural event rates in 33.3 ms bins (i.e., matching the frame rate) and weighting the frames by the event rate in the bin. A motion index was calculated by thresholding the motion map (with a threshold of 40 at 8 bit resolution) and summing the pixels exceeding the threshold. The motion index was used to define periods of quiet wakefulness and periods in which the mice were moving. Quiet periods were defined as periods in which the motion index changed by a rate of less than 0.025 arbitrary units per frame (with maximum rate of 1 given by the maximal pixel change) for at least 30 consecutive frames. Conversely, periods of movement were defined as rates of change greater than 0.025 a.u. per frame for at least 30 consecutive frames. These definitions were used to divide the corresponding electrophysiological data into periods recorded during quiet wakefulness and during locomotion for the analysis of spike and EPSC frequencies.

We correlated the binned electrophysiological activity with the motion index at either coarse (1.5 s) or fine (33 ms) time scale. The electrophysiological activity was binned as follows: For EPSCs, GC Spikes and MFT spikes, we computed the average event rate in a bin. For spillover, we computed the average spillover current (as determined by our fitting procedure below) in a bin.

For the coarse time scale correlation, we split the motion index and electrophysiological activity into 1.5 s bins and averaged the data in these bins, and then calculated Pearson's r coefficient between the event rates (for EPSCs, GC spikes, MFT spikes) or current (for spillover) and the motion index.

For the fine scale correlation, we made the bin size the duration of a video frame (i.e., 33.3 ms) for ease of analysis. We performed a rolling normalized cross-correlation between the electrophysiological data and the motion index, by shifting a 3 s window across the data. For each window, the mean was subtracted from both the motion index and electrophysiological data, and a cross-correlation performed. For each cell, an average cross-correlogram was computed by averaging across shifts, and the peak in this cross-correlogram was measured.

To establish whether these correlations were significant, we generated bootstrap samples for each cell by repeating the analysis with shuffled versions of the corresponding binned electrophysiological activity, which allowed us to generate a z-score for each cell. This z-score was used to look up a one sided p-value in a standard normal table. p-values were Bonferroni-corrected for multiple comparisons, and the significance level was set at p = 0.05.

## Analysis of spillover currents

To separate the phasic EPSCs from the underlying spillover currents we fit a model to the raw voltage clamp traces. The model consisted of a train of biexponential functions (representing the fast events) on top of a spline function, representing an underlying slow current. Formulating this mathematically, we fit a function g(t) to the data:

$$g(t) = \beta_{c_1 \ldots c_{N_{CP}}}(t) + \sum_{i=1}^{N_{EPSCs}} amp_i \times \zeta(\tau_{rise}, \tau_{fall}, t - T_i),$$

where ß is a cubic spline function with control points, $C_i$ is the ith control point of the spline, $N_{EPSCs}$ is the number of fast EPSCs in the data, $amp_i$ is the amplitude for the ith fast EPSC, $\zeta$ is the biexponential function with a rise time of $\tau_{rise}$ and a decay time $\tau_{fall}$, $T_i$ is the onset of the of the ith fast ESPC.

$$\zeta(\tau_{rise}, \tau_{fall}, t) = \frac{\left[\exp\left(-\frac{t}{\tau_{rise}}\right) - \exp\left(-\frac{t}{\tau_{fall}}\right)\right]}{(\tau_{rise} - \tau_{fall})}.$$

The data was collected in continuous sweeps of 5 s, and we therefore fit the model separately to these 5 s episodes.

The fitting procedure was as follows: a peak finding algorithm was used to detect the fast EPSCs in the raw voltage clamp traces. We initiated ß to be equal to zero, $N_{EPSCs}$ to be equal to the number of peaks found by the algorithm, $amp_i$ to the amplitudes found by the algorithms, $T_i$ the time of the events, $\tau_{rise}$ was set to 1 ms and $\tau_{fall}$ was set to 10 ms. We then used the Matlab nlinfit function to fit the model to the raw traces, adjusting the spline parameters, the $amp_i$ and $\tau_{rise}$ and $\tau_{fall}$. To correct for slow drifts in the holding current during a long recording, we set the maximum of the ß trace for each 5 s episode to be zero. The spline fits baseline shifts in the traces not accounted for by summation of fast events, and we used it as our measure of putative spillover. Note that since spillover currents are known to contribute to the tails of fast EPSCs (*DiGregorio et al., 2002*), we are underestimating the total contribution of spillover transmission in granule cells in the awake mouse.

The relative contribution of phasic and spillover transmission as a function of EPSC rate was calculated as follows: the smoothed EPSC rate was computed by convolving a causal exponential kernel (with tau = 50 ms) with a train of delta functions placed at the times of the fast EPSCs as found by our fitting procedures. We then iterated over all events in all cells and calculated the ratio of the fast current at the peak of the event on the spillover current at that time, treating each event as a data point. These data points were then binned by EPSC rate. For *Figure 2E,F* we also related the currents to the motion index, convolved with a causal exponential kernel (with tau = 660 ms).

To calculate the cross-correlation between EPSC rate and spillover (*Figure 2C*), we used the smoothed EPSC rate trace and the putative spillover traces from the fitting procedure, both mentioned above, and used a 2 s sliding window over the traces. For each window, we computed a normalized, mean subtracted cross-correlation between the EPSC rate and spillover traces. We then computed the mean cross-correlation, averaged across all cells and all the sliding windows. For the burst-triggered spillover, we defined a burst as a group of 5 or more EPSCs occurring at 200 Hz or more. We averaged the spillover trace triggered by the first event of such bursts (*Figure 2D*).

## Granule cell model

We employed a published cerebellar granule cell model (*Diwakar et al., 2009*) to study synaptic integration using in vivo patterns of activity. This model consists of a detailed compartmental model of a spiking granule cell for the NEURON simulation environment. We only modified the model by adding a fixed tonic inhibitory conductance at the soma of 1 nS ($E_{rev}$ = −70 mV). The model was run in current clamp mode. To inject the patterns of excitatory input we recorded in vivo, we added an AMPA synaptic conductance ($E_{rev}$ = 0 mV) which was varied dynamically to correspond to the

conductance underlying our voltage clamp traces, assuming a driving force of 70 mV. Our spillover analysis described above separated our voltage clamp recordings into the contributions from the fast EPSC current, and the slow putative spillover traces, so we could feed these separately or summated into the model and record the spike output of the model cell. We also performed simulations in which an NMDA receptor conductance was added to the spillover and phasic AMPA conductances, with an NMDA:AMPA ratio of 0.2 (*Cathala et al., 2003*), and a voltage-dependent $Mg^{2+}$ block modelled according to (*Nieus et al., 2006*).

## Current clamp analysis

For the current clamp GC recordings, burst analysis was done as follows: spikes were detected in the raw traces using a voltage threshold (−10 mV). A procedure then iterated through the spikes and grouped them into bursts if 3 or more spikes appeared in succession with an ISI less than 50 ms. The depolarization underlying a burst was quantified as the voltage averaged across a 5 ms interval starting 1 ms after the first spike in the burst, after subtracting a baseline voltage, defined as the average voltage over a 180 ms interval starting 20 ms before the first spike in the burst.

## PC analysis

The videos were downsampled spatially by a factor of 2, and PCs of the video were calculated by first computing the motion map of the video as above, and then shifting an 8 frame window across this motion map. For each of the videos, we created a data matrix where the rows contained the 8 frame windows of the motion maps. The standard pca function in Matlab was then used to obtain the PCs of this matrix, as well the coefficients of each video in this eigenbasis. We kept the 50 components with the highest eigenvalues for each video. For each of these components, we then computed a sliding cross-correlation between its coefficient in the video and a binned version of the electrophysiological trace (as described in the video analysis section). The sliding cross-correlation was computed as detailed above for the motion index. To establish the noise level, we generated bootstrap samples by repeating the analysis on shuffled versions of the binned electrophysiological traces.

To label the PCs according to which body parts were involved in the motion, we devised a semi-automated labelling algorithm: for each video, we defined rectangular ROIs for each of three regions (head, body, limbs). Each region could have more than one ROI (e.g., one ROI for each limb). For all the PCs, we looked for framewise pixel changes in the ROI's as follows:

$$\Delta = \sum_{x,y\in ROI} \sum_{i=2}^{n} |PC[x,y,i] - PC[x,y,i-1]|,$$

where $PC[x, y, i]$ is the PC at pixel x, y and the i-th frame, and n is the total number of frames in the PC. A ROI, and therefore a body region, was deemed to be participating in the motion of the PC if $\Delta$ exceeded a threshold. Note that in these head-fixed recordings, 'head' movement refers primarily to movement of the whisker pad and ears.

To select the threshold in a non-arbitrary manner, we also selected ROIs corresponding to background for each of the video. We then charted the distribution for the $\Delta$'s in these background ROIs. In all of the videos, $\Delta$ ranged from 0 to 2500, and therefore we selected 3000 as our threshold.

## Step analysis

We built a custom GUI to step through the videos frame by frame and annotate limb movement. We noted in each frame whether either of the forelimbs were in swing or stance phase. The swing phase was defined as starting when the paw was lifted off the ball, and ending when contact between the paw and the ball was reinitiated. For either forelimb, we calculated the step-evoked electrophysiological activity for each cell by triggering a 40-bin (or 1.32 s) episode of the activity on the start of the stance phase, and averaged these episodes for each cell, giving an average step-evoked response vector $\sigma_{1,2,...,40}$. We computed a modulation index m for each cell by taking this vector and computing:

$$m = \frac{\max_{i=1}^{40}(\sigma_i) - \min_{i=1}^{40}(\sigma_i)}{\max_{i=1}^{40}(\sigma_i) + \min_{i=1}^{40}(\sigma_i)}.$$

To test for significance, we generated bootstrap samples by selecting for each step a random window from the corresponding binned electrophysiological trace. For the individual cells, we considered a modulation index to be significant if its associated z-score corresponded to a p-value < 0.05 after Bonferroni correction for the numbers of cells within that recording modality. When measured across the population, while the mean pooled z-scores were positive, they were not significant (mean z-scores were MFBs: 2.4427 ± 2.8788, p = 0.3995 for left limb, 5.2776 ± 4.2794 for right limb. EPSCs: 3.0553 ± 1.5468, p = 0.0695 for left limb, 2.3266 ± 1.0831, 0.0522 for right limb. GC-spikes: 0.0980 ± 1.0836, p = 0.9243 for left limb, 0.1723 ± 2.0968, p = 0.9312 for right limb. Spillover: −0.1011 ± 0.3651, p = 0.7764 for left limb, 0.0192 ± 0.4422, p = 0.9645 for right limb). This is not surprising, since it reflects the fact that that only a subset of the granule cells are well-modulated by the step cycle, which is to be expected given the variety of inputs to the granule cell layer.

For the polar plot in *Figure 4—figure supplement 2*, we plotted the modulation to step cycle by taking the absolute deviation from the mean activity as the radius of the plot. That is,

$$R(\theta) = \left| activity(\theta) - \frac{\sum\limits_{i=0}^{N} activity\left(i \times \frac{360}{N}\right)}{N} \right|,$$

where activity is the step-triggered activity vector as in *Figure 4A* mapped on 360˚ (i.e., this works out to each degree in the plot corresponding to 5 ms). For ease of visualization, we subtracted from R the minimum value across phases and then divided by the maximum value, so R is between 0 and 1. We picked the phase of maximal modulation as the phase that maximized R. In *Figure 4—figure supplement 2*, we plotted the modulations as z scores by subtracting the mean of the step triggered activity vector for each recording and dividing by its standard deviation.

## HMM

A two state Markov model was constructed using the Bayes Net Toolbox (https://code.google.com/p/bnt/). The output distribution was modelled as a mixture of four Gaussians. The observed data at time i consisted of a 360 ms window of the electrophysiological data starting at time i. For the EPSCs, the activity in each window was normalized to the mean; for the spiking data, the baseline activity (average activity of the first 90 ms in the window) was subtracted. All the parameters were initiated at random, except for the prior probabilities of the states and initial state transition matrix that was derived for each cell from the step data annotated manually. The Bayes Net Toolbox was then used to perform a Baum-Welch algorithm to optimize the prior probabilities, the transmission matrix, and mixture of Gaussian parameters. The Viterbi algorithm was then used to reconstruct the predicted state transition through the HMM.

To score the prediction against the real stepping data, we first converted both the prediction and real data into spike trains, by placing spikes at the transitions from stance to swing. We then used a commonly utilized metric between spike trains (*Schreiber et al., 2003*) to compute the quality of the step reconstruction. Briefly, this measure computes a normalized, mean-subtracted cross-correlation between spike trains convolved with a Gaussian kernel (we used a sigma of 100 ms), and then takes the peak of the cross-correlation as the measure of similarity between the trains. This value can vary between 0 (no similarity) and 1 (identical trains). We multiplied this value by 100 to get a percentage score. For each limb, we took the average of the percentage score over 10 runs of the HMM algorithm to get the average prediction score. For each cell, we selected the limb that was best predicted by the activity and reported its prediction score. To test for significance for each cell, we generated bootstrap samples for each cell by repeating the above procedure on shuffled version of the electrophysiological data for the cell, and generating a z-score of the prediction score. The prediction for each cell was deemed significant if its associated one-sided p-value was <0.05 after Bonferroni correction.

## Acknowledgements

We are grateful to Tiago Branco, Arnd Roth, Christoph Schmidt-Hieber, Charlotte Arlt, and Beverley Clark for helpful discussions and for comments on the manuscript. We thank Arifa Naeem for technical assistance. This work was supported by grants from the Wellcome Trust, ERC, and Gatsby Charitable Foundation.

## Additional information

### Competing interests

MH: Reviewing editor, *eLife*. The other authors declare that no competing interests exist.

### Funding

| Funder | Grant reference | Author |
| --- | --- | --- |
| Wellcome Trust | WT086602MF | Ian Duguid |
| Wellcome Trust | 094077 | Michael Häusser |
| European Research Council (ERC) | 250345 | Michael Häusser |

The funders had no role in study design, data collection and interpretation, or the decision to submit the work for publication.

### Author contributions

KP, Conception and design, Acquisition of data, Analysis and interpretation of data, Drafting or revising the article; AM, ID, Conception and design, Acquisition of data, Analysis and interpretation of data, Drafting or revising the article, Contributed unpublished essential data or reagents; MH, Conception and design, Analysis and interpretation of data, Drafting or revising the article

### Ethics

Animal experimentation: This study was performed in strict accordance with UK Home Office regulations. Experiments were carried out under Project Licence 70/7833 issued by the Home Office, which was issued following local ethical review, and under the relevant Personal Licences issued by the Home Office. All surgery was performed under isoflurane anesthesia, and every effort was made to minimize suffering.

## Additional files

### Supplementary file

• Source code 1. Custom built software in Matlab (Mathworks).

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
