## [Decision Letter]

Thank you for sending your work entitled “Synaptic representation of locomotion in single cerebellar granule cells” for consideration at *eLife*. Your article has been favorably evaluated by Eve Marder (Senior editor), a Reviewing editor, and three reviewers.

The Reviewing editor and the reviewers discussed their comments before we reached this decision, and the Reviewing editor has assembled the following comments to help you prepare a revised submission.

The primary issues (from the reviewers' substantive comments below) relate to reanalyzing some data, further analyzing other data, and providing more complete descriptions of methods to address the major points of the reviewers by doing the following:

1) Examining or discussing the effect of inhibition on granule cells during locomotion [Reviewer 1, point 3];

2) More clearly distinguishing the role of spillover relative to direct transmission and/or validating conclusions made [Reviewer 1, point 4];

3) Providing more detail on recording parameters, mouse motion states, principal components, and lateralization [Reviewer 2; Reviewer 3, point 2];

4) Considering the physical referents of the model more carefully, regarding whether the granule cells are signaling sensory vs. motor information [Reviewer 3, point 1].

The reviewers have raised a number of additional points that will also require clarification. The full set of comments is attached below to facilitate your revisions.

*Reviewer #1*:

In this manuscript Powell et al. studied the synaptic inputs into cerebellar granule cells during locomotion. Using in vivo patch-clamp recordings, authors illustrated a correlation between locomotion and neural activities at the input stage of cerebellum, i.e. increasing mossy fiber and granule cell activities, and the increased spiking in granule cells is facilitated by glutamate spillover currents. Furthermore, Powell et al. found that the locomotion circle can be precisely predicted by the activities of some granule cells and suggested an alternative interpretation of information transfer from granule cells, which differs from the classical view of sparse coding.

This study aims to illustrate an important question in the cerebellar research (how do the cerebellar neurons encode information during locomotion), using challenging techniques (whole cell recordings in awake locomoting mice). The results are interesting and presented well. I have several suggestions to further improve the manuscript.

Major points:

1) It is somewhat surprising that most MF boutons and all granule cells increased their spiking firing during locomotion. As the mossy fiber inputs encode multiple modalities, and these modalities can converge onto the same granule cells, it remains unclear how come that all the recorded granule cells increased firing during locomotion. For instance, the increase of mossy fiber activity in Figure 1 shows a clear lag behind the onset of locomotion and a prolonged period after the termination of walking. This raises the question as to what extent the neural activities are influenced by the shift in brain state during locomotion?

2) In the last sentence of the subsection headed “Whole-cell recordings from granule cells and mossy fibers during locomotion”, what do the authors mean by ‘a more dense code’? I don't think a difference in CV value can be translated into the difference in coding density. Since the previous work from authors' lab suggested a high feudality, detonator-like information transfer onto granule cells, it is unclear to me how this difference in coding density came about.

3) What is the impact of synaptic inhibition on granule cells during locomotion? MF inputs can also influence the granule cell output via the inhibitory inputs from Golgi cells, which is thought to create a proper time window for granule cell spiking (13). The IPSC pattern during locomotion should also be examined.

4) The importance of spill-over during locomotion is unconvincing due to the lack of proper experimental data. Indeed, the behavioural analysis showed that the spiking patterns are correlated to locomotion, rather than the occurrence of spill-over transmission per se. The computational model in Figure 2 can be simply explained by two sub-threshold inputs summating to create a supra-threshold output. Given a supra-threshold conductance, can spill-over or phasic input alone create supralinear spike output; this needs to be examined.

5) In the quiet state, mossy fiber boutons (MFB) were found to fire at around 12 Hz (as detailed in the subsection “Whole-cell recordings from granule cells and mossy fibers during locomotion”). Under those circumstances, granule cells showed EPSCs at approx. 78 Hz (almost 7x as often). When active, the MFB fired around 46 Hz, while granule cells had EPSCs at ∼114 Hz (almost 2.5x more often). Given that in each granule cell receives input from approx. 4 MFBs, independent of the behavioural state, one would expect a ratio of approx. 1:4 (as reported in the first paragraph of the above mentioned subsection). Please discuss this in more detail.

6) The report of spiking frequency of granule cells, especially during locomotion is a bit confusing. The authors report an average rate of 5 Hz, but burst of up to over 100 Hz (in the second paragraph of the subsection headed “Whole-cell recordings from granule cells and mossy fibers during locomotion”, Figure 1). Burst firing is ill described by using the mean firing rate.

7) What do the authors mean with “anatomically coherent regions” (subsection “Linking synaptic input parameters with output spiking during locomotion”)? Please give more details here.

8) The authors argue, and this seems plausible, that the neuronal firing patterns correspond to different features of motion rather than to the degree/intensity/amplitude of each motion. Can the authors elaborate on this? Does this imply that granule cells do not encode the degree/intensity/amplitude of each motion parameter? Or do they do both? Please provide in detail how many cells, what percentage of the cells show what, etc.

9) Last sentence in the Discussion. I don't think authors' data provide sufficient argument against the sparse coding in cerebellum. As mentioned previously, it is unclear what the extremely high incidence of detecting responsive granule cells really means. If this probability reflexes the true population response, that must indicate that almost all granule cells in lobule 5 are encoding for similar information during locomotion. If so, what do the authors mean by ‘a population of robustly tuned neurons’?

*Reviewer #2*:

This is a potentially interesting study that provides new results on information processing at the input stage of the cerebellar cortex during behavior. A strong point of the paper is that the authors use a variety of experimental and modeling approaches to address the issue. My main problem is that many of the methods are results are not described in enough detail to fully judge the validity of the findings and the strength of the conclusions.

Results section: the durations of the various recordings (MFs and granule cells) is not given. The averages and minimum acceptable recording durations should be given for the various conditions.

The definitions of resting versus quite wakefulness need to be made clear. Otherwise the various firing rate values seem inconsistent. For example, at rest MF spikes and granule cell EPSCs occurred at 21 and 60 Hz. But during quiet rest the values were 11.7 and 77. 5 Hz (note the divergence between the spikes and EPSCs). So, how do the two conditions relate to each other and to the locomotion state?

In the first paragraph of the subsection headed “Whole-cell recordings from granule cells and mossy fibers during locomotion”: It is stated that the MF frequency of 21 Hz and the EPSC frequency of 60 Hz are consistent with there being 4 MF inputs. However, it seems that the ratio is more consistent with a granule cell having only 3 MF inputs, which doesn't fit with the anatomy. How can you explain the discrepancy?

In the subsection entitled “Whole-cell recordings from granule cells and mossy fibers during locomotion”: A motion index was used to categorize the state of the animal as being either in quiet wakefulness or voluntary locomotion. From the description of the motion index in the Methods, one can perhaps see that it indicates when the animal is making movements, but it is hard to see how it shows that the animal is specifically locomoting.

Further the description of the BG calculation is vague. How long is ‘a long time’? The nature of the objects in the formulas is not entirely clear to me. *BG*, *Frame* and *im* seem to be matrices but *diffim* seems to be a single number based on the summation sign. Is BG the same size as the entire Frame? The calculation is not described consistently between the Methods and the legend for Figure 1: in the Methods the motion index is described as a summation, but in the legend it is described as a fraction.

In the second paragraph of the subsection “Whole-cell recordings from granule cells and mossy fibers during locomotion”: It seems that only 4 MFs were recorded for the comparison between quiet rest and ‘locomotion’. This seems a very low number to base a conclusion on, particularly when 3 MFs increased activity, and 1 decreased.

In the third paragraph of the subsection headed “Glutamate spillover currents drive spiking during locomotion”: The correlation r=.44 in the text doesn't match the r=.45 given in the legend for Figure 2. Also, with an n=7, the p-value doesn't match what is given (at least for a t-test). Thus, I assume the n for the statistical test is actually the number of bursts shown in the Figure 2 (about 35-40). If this is the case, it suggests that very few bursts were examined in each cell (about 5). Is this correct? Given the distribution of the residuals about the regression line in Figure 2 one wonders if the relationship varied considerably between recordings. Is a relationship seen on an individual cell basis?

How were EPSCs counted? From Figure 1 they appear to have many amplitudes, suggesting summation of synchronous events. However, in the Methods it appears that a simple single level (2xSD) was used. This could significantly under count EPSCs particularly for the high frequency conditions.

In the second paragraph of the subsection headed “Linking synaptic input parameters with output spiking during locomotion”: It is not clear which of the principal components ended up in the top 50 or top 10. Did these correspond to limb, head, rest of body motions and/or combinations of these three motions. Even if manual inspection was used, some criteria must have been used and should be described. Were the decisions made blindly by several people to test for consistency across observers?

*Reviewer #3*:

In this manuscript, Powell et al. performed in vivo whole-cell patch-clamp recordings from cerebellar granule cells (GCs) and mossy fibre (MF) terminals in awake mice during locomotion. They investigated how synaptic inputs and spike outputs of GCs represented gait information. They first found that frequencies of both inputs and outputs of GCs increased during movements of mice and also showed that putative spillover currents had significant contribution to spike outputs during movements. The authors next showed that MF spikes, GC EPSCs, GC spillover and GC spikes well correlated with movements. Finally, they succeeded in reconstructing gait patterns from input and output patterns of single GCs.

The data provided are in very high quality as we usually expect from this group and I congratulate them on the considerable technical achievements made in this study. There is no question that information during locomotion is conveyed by either synaptic inputs or spike outputs of GCs. The key question is whether movement parameters are indeed represented by the activities of GCs that they observed, which is the main claim here. Unfortunately, I am not fully convinced with the interpretation of the data and the significance of the findings. I have the following concerns that need to be addressed.

1) In Figure 4, the authors performed the decoding of input and output patterns of GCs using Hidden Markov Model, and successfully reconstructed gait patterns. However, it does not necessarily mean that these signals contain information of movements given that the two-state model they used may only represents sensory inputs (touched and untouched) rather than movements (stance and swing). Since they filmed the movements of forelimbs, they should try to track the trajectories of forelimb movements and then reconstruct them from GC activities.

2) In Figure 2, glutamate spillover currents significantly contributed to the spike outputs of GCs, and subthreshold depolarization strongly correlated with the frequency of burst firing of GCs during movements (Figure 2). However in Figures 3 and 4, peak correlation between the motion index and the spillover was fairly low. The authors explain that this is due to slow time course of spillover currents (subsection “Linking synaptic input parameters with output spiking during locomotion”). However, the strong correlation between spillover currents and GC spike outputs indicates that the time course of change in CG spike rate is similar to that of spillover currents (Figure 2, left). Therefore the correlation between spillover currents and the motion index is expected to be similar to that of GC spikes.

3) Analyses of the difference between the left and right limbs are interesting. In particular, the direction of modulation seems to be opposite in a single GC (Figure 4). It is a pity that the authors only analyzed the absolute depth of modulation and discarded the information about alternation of the left and right limbs. They should analyze the relation between the left-right alternation and GC activities, which should also be related to the laterality of the recording site.

[Editors' note: further revisions were requested prior to acceptance, as described below.]

Thank you for resubmitting your work entitled “Synaptic representation of locomotion in single cerebellar granule cells” for further consideration at *eLife*. Your revised article has been favorably evaluated by Eve Marder (Senior editor), a Reviewing editor, and three reviewers. The manuscript is greatly improved but there are some remaining issues that need to be addressed before acceptance, as outlined below:

The primary issue is whether the data support the conclusion that the granule cell activity is really correlated to “specific attributes” of the motion, as detailed in Reviewer 2's point #2 below. This issue could be most simply addressed by editing the text to curtail the extent of the conclusions.

*Reviewer #1*:

The authors have addressed virtually all issues. One relevant point left, regarding reviewer 1's major point 6: the authors improved their description of the spiking patterns observed, as they now explicitly refer to particularities of burst firing (in the third paragraph of the subsection headed “Whole-cell recordings from granule cells and mossy fibers during locomotion”). Unfortunately, in Figure 1, the authors still present a trace with an instantaneous firing rate of probably well above 100 Hz (Figure 1) next to a summary stating a maximum frequency of around 15 Hz (Figure 1). I think it would be better to address the burst firing also in the figure itself. All the rest is well done.

*Reviewer #2*:

The authors have clarified and addressed the points previously raised. There are some significant findings. Most significant is the recording of granule cell activity in awake animals and the comparison of activity during restful waking and locomotion. The major remaining concern I have is with the interpretation of the results. I think given the low sample sizes for the locomotion recordings (particularly for the mossy fibers, but even for the granule cells) the authors need to be conservative in their conclusions.

1) In first lines of the Discussion and elsewhere the authors imply that they have “directly measured the transformation from synaptic input patterns to spike output in single neurons”. This statement suggests that they have measured the inputs and outputs to the same cell and that they have defined the transform for individual cells. They have not. What they have done is record two populations of cells and try to infer how activity transforms from one to the other. The problem is that each of these populations is heterogeneous in terms of spontaneous and behavior-related activity. Thus, any number of transforms is possible depending on how MFs connect with granule cells. Even the recordings of granule cell EPSCs and spikes were done for different (though perhaps overlapping) populations of cells, so we cannot say how the EPSCs relate to the specific firing patterns.

2) The authors state that their results show that the activity of individual granule cells is tuned to “specific features of the animal's motion, rather than merely correlating with just the amount of motion”, and that they can reconstruct the “entire step sequence” (subsection “Linking synaptic input parameters with output spiking during locomotion”). But it seems that their results are more consistent with a general association between changes in granule cell activity and movement.

First, the analysis of the video shows good correlations between movement (motion index) and neuronal activity when large time windows (bins of 1.5 s duration) are used but much poorer correlations when “high” resolution windows of 33 ms, which themselves are actually not particularly precise in terms of motor control, are used. Compare Figure 3 but note the different scales of the y axes.

Second, Figure 3—figure supplement 1 shows that the increase in neuronal activity begins 100's of milliseconds ahead of the actual movement and decreases to baseline over the course of a second following the cessation of movement. This lack of precise temporal linking of neuronal activity and locomotion is also apparent in Figure 1.

Third, the principle components derived from the video (and used to correlate with neuronal activity) do not seem to be reflecting particularly specific features, at least based on their descriptions. For example, the individual components were said to represent limb, head, and “rest of body” motion. Moreover, the sum of the components representing each of the three categories is 1542, meaning that many of the 1000 components represent more than one category of movement.

Fourth, while the analyses shown in Figure 4 certainly show that periods of movements predicted by the neuronal activity generally match up with the actual periods of movement, it does not appear that the individual predicted movements align particularly well with the actual movements. Moreover, for most of neuronal signals only a small minority of cells showed a significant predictive quality, and none showed a predictive quality value approaching 100%, which would have implied a matching of the entire step sequence. Furthermore, the predictive quality measure was only used to look at stance-to-swing transitions. One wonders why the swing-to-stance transitions were not similarly analyzed.

*Reviewer #3*:

The authors have addressed most of this reviewer's comments satisfactorily and have improved their manuscript significantly. The topics of this manuscript are very important for the researchers in this field and also of interest for neuroscientists in general. I therefore recommend acceptance of this manuscript for publication in *eLife*.

---

## [Author Response]

*The primary issues (from the reviewers' substantive comments below) relate to reanalyzing some data, further analyzing other data, and providing more complete descriptions of methods to address the major points of the Reviewers by doing the following*:

*1) Examining or discussing the effect of inhibition on granule cells during locomotion [Reviewer 1, point 3]*;

Although our manuscript has focused on excitation of granule cells during locomotion, we agree that this is an interesting issue. We have obtained a recording of a granule cell at an intermediate holding potential where both IPSCs and EPSCs are detectable. In this recording, which is shown in Figure 5, the cross-correlation between activity and the motion index is much lower for the IPSCs than EPSCs. This echoes results in other sensory areas showing that tuning of inhibition is broader than for excitation. While this data is interesting and provocative, the extremely low experimental yield for these experiments means the role of inhibition would be better explored in a subsequent project dedicated to this question. We have now included a discussion of the effects of inhibition on granule cells during locomotion, as suggested.

*2) More clearly distinguishing the role of spillover relative to direct transmission and/or validating conclusions made [Reviewer 1, point 4]*;

The reviewers requested a clarification of the role of spillover versus direct transmission. Our argument is that while spillover and direct (fast) transmission are correlated, spillover transmission provides mainly a slow increase in excitability, whereas the direct transmission provides fine-grained information about the animal’s motion/sensory state. Reviewer 1 is correct that spillover and direct currents interact to generate additional output spikes. However, we demonstrate that what the spillover current is doing is effectively providing an independent means of regulating the synaptic gain. To explore this, we have now provided a new figure, which plots the total synaptic charge as a function of EPSC rate with and without spillover (Figure 2—figure supplement 1). This demonstrates that spillover is a crucial determinant of synaptic gain. Moreover, the model results illustrated in Figure 2 show that this gain control provided by spillover is in a range that is physiologically relevant, i.e. it can have a major impact on spiking output by the granule cell.

Reviewer 3 is concerned that despite the high correlations (Figure 2) between spillover and fast transmission, the modulation of spillover by locomotion (Figure 4) is lower than for fast transmission. To illustrate what we think is going on, we have made a new supplementary figure (Figure 4—figure supplement 1) in which we show that spillover acts like a smoothing temporal filter. We have taken the bins representing the EPSCs rate in our cells (n = 9), and convolved them with a biexponential kernel with rise time 50 ms and decay time 100 ms (these values were selected to mimic the on-off kinetics of glutamate spillover, see for instance Mitchell and Silver, Nature 2000); please note that the precise values are not important for the qualitative point we are making. Cross-correlating this filtered signal with the original EPSCs signal gives a qualitatively similar cross-correlation as with the actual spillover recorded (compare with Figure 2), with a high peak value of 0.5. However, if we compare the locomotion modulation index of the filtered signal with the original EPSCs, we see that the filtered signal tracks the animal’s step much less well (Figure 4—figure supplement 1) – effectively because it is low pass filtering the information at crucial frequencies.

In summary, we believe our data provides strong evidence that spillover is important for regulating the excitability of the granule cell to enhance the transmission of information during locomotion. To our knowledge, ours is the first study to show that spillover transmission is engaged in vivo during behaviour in this way.

*3) Providing more detail on recording parameters, mouse motion states, principal components, and lateralization [Reviewer 2; Reviewer 3, point 2]*;

To address these issues we have done the following:

Added a figure on the lateralization of the step cycle modulation (Figure 4—figure supplement 2). Here we show, by drawing an analogy with orientation tuning in the visual system, that granule cell responses can show modulations at different phases of the step cycle, and that comparing the maximal modulation of one limb with the other shows that these peak modulations tend to happen at around 90 degrees. The direction of modulation (up or down) can vary between cells.

We have improved our classification of principal components using a semi-automated labelling algorithm, now labelling all 1000 of the principal components. We have provided more details about the classification of principal components in the Methods.

We have added a figure showing that the relationship between EPSC rates goes up at locomotion onset, and down on locomotion termination (Figure 3—figure supplement 1).

*4) Considering the physical referents of the model more carefully, regarding whether the granule cells are signaling sensory vs. motor information [Reviewer 3, point 1]*.

This is an important issue, but it is challenging to reliably distinguish whether the granule cell signals we record are sensory or motor given that the whole-body movement associated with locomotion will engage a range of sensory modalities and motor programs. Given what is known about the origin of mossy fibers projecting to this region of cerebellum, it is likely that sensory and proprioceptive information related to movement at least partially underlies the responses we observe, and we have discussed this in the manuscript. Furthermore, given the sustained increased response rates after locomotion onset (Figure 3—figure supplement 1), we agree with reviewer 1 that there does appear to be a general increase in rate due to movement-related brain states. However, these would not account for the step cycle modulations. In general, we would like to suggest that our paradigm represents an excellent example of sensory-motor integration, in which both sensory and motor information must be combined for optimal performance. Dissection of distinct sensory and motor components will require a substantial additional study (e.g. involving optogenetic tagging and/or silencing of mossy fiber inputs from different origins).

In summary, we are confident that we have addressed all of the comments of the editors and reviewers. These changes have substantially strengthened the manuscript. A detailed response to each of the comments of the referees is provided below.

Reviewer #1:

*1) It is somewhat surprising that most MF boutons and all granule cells increased their spiking firing during locomotion. As the mossy fiber inputs encode multiple modalities, and these modalities can converge onto the same granule cells, it remains unclear how come that all the recorded granule cells increased firing during locomotion. For instance, the increase of mossy fiber activity in*
Figure 1
*shows a clear lag behind the onset of locomotion and a prolonged period after the termination of walking. This raises the question as to what extent the neural activities are influenced by the shift in brain state during locomotion*?

The referee has raised a number of important issues. We would like to make several points in response. First, although all granule cells increase their firing rate during locomotion, the relative increases in firing rate span a wide range. The fact that the firing rates of some granule cells only increase very slightly indicates that the granule cells we recorded represent a diverse population with respect to their selectivity for different movement or sensory parameters. Second, please note that Figure 1 shows increases associated with all motion—locomotion as well as grooming, etc. To address this, we have now added a new figure (Figure 3—figure supplement 1) which shows EPSC rates just before and after periods of locomotion. Interestingly, EPSC activity does seem to “ramp up” before the steps start, and be sustained longer than the last step. However to make a precise statement about the timing relation of this activity to total motor activity would require a more sensitive measure such as EMG over several muscle groups, which would necessitate a different experimental approach (and thus should be done in a subsequent study). Third, we do agree that there appears to be a shift in brain state during locomotion (which is hardly surprising given that this type of whole-body movement is likely to engage many sensory modalities and motor programs). However, the short-term modulations in event rates we explore in Figures 3 and 4 cannot be accounted for only by this state shift, since we are correcting for the average event rate (by using normalized cross-correlations and a modulation index in which the mean activity is corrected for). Finally, what this discussion illustrates is that the remarkable sensitivity and temporal resolution of these recordings offer an unprecedented view into how individual elements of the granule cell population parse information about sensory-motor integration, and set the stage for a range of different avenues to be explored in future using this approach.

*2) In the last sentence of the subsection headed “Whole-cell recordings from granule cells and mossy fibers during locomotion”, what do the authors mean by ‘a more dense code’? I don't think a difference in CV value can be translated into the difference in coding density. Since the previous work from authors' lab suggested a high feudality, detonator-like information transfer onto granule cells, it is unclear to me how this difference in coding density came about*.

The key result is that there is a massive increase in synaptic input to the granule cell layer during locomotion—equivalent to a state change of the system—which in turn raises the firing rate of the granule cells by an order of magnitude. This level of activity in the network fits the standard definition of ‘dense coding’ (activity density of ∼0.5; see Foldiak & Endres, http://www.scholarpedia.org/article/Sparse_coding). This difference in coding density appears to be primarily related to an increase in the fraction of presynaptic mossy fibers that are active. However, we agree with the referee that a difference in CV cannot be directly translated into a difference in coding density, and have now removed this sentence.

*3) What is the impact of synaptic inhibition on granule cells during locomotion? MF inputs can also influence the granule cell output via the inhibitory inputs from Golgi cells, which is thought to create a proper time window for granule cell spiking (*[13]*). The IPSC pattern during locomotion should also be examined*.

This is an interesting issue, but extremely challenging to address directly given the difficulty of these experiments (and in particular experiments designed to record inhibition, which are notoriously challenging). We have obtained a recording of a granule cell at an intermediate holding potential where both IPSCs and EPSCs are detectable. In this recording, which is shown in Figure 5, the cross-correlation between activity and the motion index is much lower for the IPSCs than EPSCs. This appears to echo results in other sensory areas showing that tuning of inhibition is broader than for excitation (reviewed by Isaacson and Scanziani, Neuron 2011). While this data is interesting and provocative, the extremely low experimental yield for these experiments means the role of inhibition would be better explored in a subsequent project dedicated to this question. We have now included a discussion of the effects of inhibition on granule cells during locomotion, and cited the D’Angelo and De Zeeuw review, as suggested.

Author response image 1.Simultaneous recording of IPSCs and EPSCs during locomotion.(A) recording of EPSCs (inward currents) and IPSCs (outward currents) concurrently in a granule cell voltage clamped at ∼-30 mV. (B) Cross-correlation with motion index for IPSCs (green) and EPSCs (red), shows a much stronger cross-correlation for EPSCs than IPSCs.**DOI:**
http://dx.doi.org/10.7554/eLife.07290.014

*4) The importance of spill-over during locomotion is unconvincing due to the lack of proper experimental data. Indeed, the behavioural analysis showed that the spiking patterns are correlated to locomotion, rather than the occurrence of spill-over transmission per se. The computational model in*
Figure 2
*can be simply explained by two sub-threshold inputs summating to create a supra-threshold output. Given a supra-threshold conductance, can spill-over or phasic input alone create supralinear spike output; this needs to be examined*.

Reviewer 1 is correct that spillover and direct currents interact to generate additional output spikes. However, we demonstrate that what the spillover current is doing is effectively providing an independent means of regulating the synaptic gain. To explore this, we have now provided a new figure, which plots the total synaptic charge as a function of EPSC rate with and without spillover (Figure 2—figure supplement 1). This demonstrates that spillover is a crucial determinant of synaptic gain. Moreover, the model results illustrated in Figure 2 show that this gain control provided by spillover is in a range that is physiologically relevant, i.e. it can have a major impact on spiking output by the granule cell.

*5) In the quiet state, mossy fiber boutons (MFB) were found to fire at around 12 Hz (as detailed in the subsection “Whole-cell recordings from granule cells and mossy fibers during locomotion”). Under those circumstances, granule cells showed EPSCs at approx. 78 Hz (almost 7x as often). When active, the MFB fired around 46 Hz, while granule cells had EPSCs at ∼114 Hz (almost 2.5x more often). Given that in each granule cell receives input from approx. 4 MFBs, independent of the behavioural state, one would expect a ratio of approx. 1:4* (as reported in the first paragraph of the above mentioned subsection*). Please discuss this in more detail*.

The reviewer is concerned that the ratio of the GC EPSC rate to MFT spike rate is not consistent with the 4:1 convergence of mossy fiber terminals per granule cell. However the reviewer is dividing the mean EPSC rate by the mean spike rate, which is statistically unwarranted (especially for low n). Instead, one needs to calculate confidence intervals for the ratio of the mean event rates. There are several approaches that can be used (see for instance [35]). For our data, we have employed the bootstrap method. In short, this creates a bootstrap distribution for the ratio by resampling with replacement of the MFT and EPSC event rates. Using this method, we get a 95% confidence interval of 3.49 to 17.67 at rest, and 1.69 to 8.39 during motion. Both of these confidence intervals are consistent with the 4 to 1 convergence.

*6) The report of spiking frequency of granule cells, especially during locomotion is a bit confusing. The authors report an average rate of 5 Hz, but burst of up to over 100 Hz (in the second paragraph of the subsection headed “Whole-cell recordings from granule cells and mossy fibers during locomotion”,*
Figure 1*). Burst firing is ill described by using the mean firing rate*.

We have performed additional analysis to address this point. 4 out of 7 cells fired in bursts (defined as groupings of 4 or more spikes with ISIs less than 50 ms). These bursts, containing an average of 11.9 ± 2.1 spikes with ISI 10.7 ± 3 ms, occurred with an average inter-burst interval 1.88s ± 0.99 s. These numbers have been added to the manuscript.

*7) What do the authors mean with “anatomically coherent regions” (subsection “Linking synaptic input parameters with output spiking during locomotion”)? Please give more details here*.

We mean that there were confluent regions of pixels that correspond to either the mouse's limb, head, body and/or tail. For instance in Figure 3, the spike triggered average shows a region of high signal which clearly corresponds to the left forelimb. In our principal component analysis, we have now improved our analysis of the principal components to describe all the 1000 principal components in a more objective way.

*8) The authors argue, and this seems plausible, that the neuronal firing patterns correspond to different features of motion rather than to the degree/intensity/amplitude of each motion. Can the authors elaborate on this? Does this imply that granule cells do not encode the degree/intensity/amplitude of each motion parameter? Or do they do both? Please provide in detail how many cells, what percentage of the cells show what, etc*.

What we mean is that the spiking seems to correlate better with specific features of the motions rather than the total amount of motion (because we get higher correlations with specific principal components than with the motion index). Our new analysis (Figure 4—figure supplement 2), looking at “tuning” to the step cycle corroborates this. To further address this in detail, subsequent studies are needed to correlate a detailed kinematic map of the mouse's motion with the neural activity pattern; unfortunately this is beyond our capabilities at the moment.

*9) Last sentence in the Discussion. I don't think authors' data provide sufficient argument against the sparse coding in cerebellum. As mentioned previously, it is unclear what the extremely high incidence of detecting responsive granule cells really means. If this probability reflexes the true population response, that must indicate that almost all granule cells in lobule 5 are encoding for similar information during locomotion. If so, what do the authors mean by ‘a population of robustly tuned neurons’*?

First, as discussed above, different granule cells show a wide range of increases in firing rate associated with locomotion. Second, the new Figure 4—figure supplement 2 shows that neurons are tuned to specific phases of the gait cycle and don’t all encode the same thing, so it is not a non-specific increase in activity. Rather, our data provide support for the idea that the representation switches from sparse (i.e. a small fraction of granule cells active) to dense (a large fraction of granule cells active), with the information conveyed by modulations in spike rates, which does not rule out that there may be timing codes embedded in the dense activity (as suggested by the phase-specificity).

Reviewer #2:

*This is a potentially interesting study that provides new results on information processing at the input stage of the cerebellar cortex during behavior. A strong point of the paper is that the authors use a variety of experimental and modeling approaches to address the issue. My main problem is that many of the methods are results are not described in enough detail to fully judge the validity of the findings and the strength of the conclusions*.

*Results section: the durations of the various recordings (MFs and granule cells) is not given. The averages and minimum acceptable recording durations should be given for the various conditions*.

The durations of the recordings have now been included in the manuscript.

*The definitions of resting versus quite wakefulness need to be made clear. Otherwise the various firing rate values seem inconsistent. For example, at rest MF spikes and granule cell EPSCs occurred at 21 and 60 Hz. But during quiet rest the values were 11.7 and 77. 5 Hz (note the divergence between the spikes and EPSCs). So, how do the two conditions relate to each other and to the locomotion state*?

The two conditions were always defined in the same way. Quiet periods were defined as periods in which the motion index was changing by a rate of less than 0.025 arbitrary units per frame (with maximum rate of 1 defined from the maximum pixel changes) for at least 30 consecutive frames. Periods of movement were defined as rates of change greater than 0.025 a.u. per frame for at least 30 consecutive frames. These definitions were used to grossly divide the electrophysiological data into periods with and without movement. In some mice voluntary movement was not observed during the recording, but this data was included when calculating overall event frequencies for the different cells types during quiet wakefulness (subsection “Whole-cell recordings from granule cells and mossy fibers during locomotion”). In a subset of cells, activity was recorded during periods of locomotion and quiet wakefulness. In this subset of cells we directly compared the activity recorded during these two conditions. The text has been rewritten to clarify these points.

*In the first paragraph of the subsection headed “Whole-cell recordings from granule cells and mossy fibers during locomotion”: It is stated that the MF frequency of 21 Hz and the EPSC frequency of 60 Hz are consistent with there being 4 MF inputs. However, it seems that the ratio is more consistent with a granule cell having only 3 MF inputs, which doesn't fit with the anatomy. How can you explain the discrepancy*?

The reviewer is concerned that the ratio of the GC EPSC rate to MFT spike rate is not consistent with the 4:1 convergence of mossy fiber terminals per granule cell. However, this is based on simply dividing the mean EPSC rate by the mean spike rate, which is statistically unwarranted (especially for low n). Instead, one needs to calculate confidence intervals for the ratio of the mean event rates. There are several approaches that can be used (see for instance [35]). For our data, we have employed the bootstrap method. Briefly, this creates a bootstrap distribution for the ratio by resampling with replacement of the MFT and EPSC event rates. Using this method, we get a 95% confidence interval of 3.49 to 17.67 at rest, and 1.69 to 8.39 during motion. Both of these confidence intervals are consistent with the 4 to 1 convergence.

*In the subsection entitled “Whole-cell recordings from granule cells and mossy fibers during locomotion”: A motion index was used to categorize the state of the animal as being either in quiet wakefulness or voluntary locomotion. From the description of the motion index in the Methods, one can perhaps see that it indicates when the animal is making movements, but it is hard to see how it shows that the animal is specifically locomoting*.

The motion index used for Figures 1 and 2 measures all movements. These are primarily locomotion-related, but can also include movements associated with grooming and other behaviours. Figures 3 and 4, and the new supplementary figures, look specifically at locomotion. We have revised the text to clarify this.

*Further the description of the BG calculation is vague. How long is ‘a long time’? The nature of the objects in the formulas is not entirely clear to me.* BG*,* Frame *and* im *seem to be matrices but* diffim *seems to be a single number based on the summation sign. Is BG the same size as the entire Frame? The calculation is not described consistently between the Methods and the legend for*
Figure 1*: in the Methods the motion index is described as a summation, but in the legend it is described as a fraction*.

Our background subtraction algorithm is a variant on common background subtraction techniques (e.g. see M. Piccardi, Background subtraction techniques: a review. Systems, man and cybernetics, 2004 IEEE international conference, Vol. 4. IEEE, 2004). We have changed the wording of the Methods section to be clearer about which of the values are vectors or scalars. With our alpha of 0.3, this means that the background is averaging over about 15 frames, which is about 500 ms. The method calculates an iterative weighted average of the current frame with the previous background, so that past frames contribute to the current background estimate with decaying weights. We have graphed this weight decay function below (see Figure 6).

Author response image 2.The contribution of the frame to the current background estimate as a function of the distance of the past frame to the current frame. See Materials and methods for a detailed description.**DOI:**
http://dx.doi.org/10.7554/eLife.07290.015

In Figure 1 in the manuscript, we show the motion index normalized to the peak value in the whole session for simplicity. However, we stress that the actual units of the motion index are irrelevant for the correlational analyses, because we use normalized cross-correlations.

*In the second paragraph of the subsection “Whole-cell recordings from granule cells and mossy fibers during locomotion”: It seems that only 4 MFs were recorded for the comparison between quiet rest and ‘locomotion’. This seems a very low number to base a conclusion on, particularly when 3 MFs increased activity, and 1 decreased*.

We agree that on their own these numbers would make a weak basis for the conclusion. However, in conjunction with the fact the EPSCs go in the same direction (since the EPSCs originate from the mossy fibers), we remain confident that this is a robust finding.

*In the third paragraph of the subsection headed “Glutamate spillover currents drive spiking during locomotion”: The correlation r=.44 in the text doesn't match the r=.45 given in the legend for*
Figure 2*. Also, with an n=7, the p-value doesn't match what is given (at least for a t-test). Thus, I assume the n for the statistical test is actually the number of bursts shown in the*
Figure 2
*(about 35-40). If this is the case, it suggests that very few bursts were examined in each cell (about 5). Is this correct? Given the distribution of the residuals about the regression line in*
Figure 2
*one wonders if the relationship varied considerably between recordings. Is a relationship seen on an individual cell basis*?

We apologize for this typo and have fixed the text and legend. The P-value is indeed across the bursts, not cells. We have also made the text more precise: out of the 7 cells, 4 had bursts, adding up to 37 bursts in total. The figure below relates the burst depolarization with the burst frequency—there is a good correlation (Figure 7). There are too few bursts per cell to comment on whether this population trend is also visible at the single cell level.

Author response image 3.Relationship between burst depolarization and burst frequency. Each colour represents a different cell from which the burst was recorded.**DOI:**
http://dx.doi.org/10.7554/eLife.07290.016

*How were EPSCs counted? From*
Figure 1
*they appear to have many amplitudes, suggesting summation of synchronous events. However, in the Methods it appears that a simple single level (2xSD) was used. This could significantly under count EPSCs particularly for the high frequency conditions*.

The event detection algorithm (together with careful manual inspection of all the data) allows detection of events with very small inter-event intervals (∼0.2 ms). This allows even summating EPSCs to be counted as individual events.

*In the second paragraph of the subsection headed “Linking synaptic input parameters with output spiking during locomotion”: It is not clear which of the principal components ended up in the top 50 or top 10. Did these correspond to limb, head, rest of body motions and/or combinations of these three motions. Even if manual inspection was used, some criteria must have been used and should be described. Were the decisions made blindly by several people to test for consistency across observers*?

We have addressed this concern by substituting our manual labelling of the principal components with a more objective semi-automated method. This has allowed us to label all the principal components rather than a few. The method is described in detail in the paper, but briefly we defined regions of interest (ROIs) in the videos corresponding to the body parts and measured the pixel changes in these ROIs for all the principal components, and decided based on a threshold what body parts were involved in the principal component. Note that the proportions reported are in keeping with our initial manual labelling.

Reviewer #3:

*1) In*
Figure 4*, the authors performed the decoding of input and output patterns of GCs using Hidden Markov Model, and successfully reconstructed gait patterns. However, it does not necessarily mean that these signals contain information of movements given that the two-state model they used may only represents sensory inputs (touched and untouched) rather than movements (stance and swing). Since they filmed the movements of forelimbs, they should try to track the trajectories of forelimb movements and then reconstruct them from GC activities*.

Our video recordings don't allow accurate tracking of the forelimb trajectories. However we think it will be very fruitful for further studies to build a proper 3D kinematic model of the limbs during locomotion, and use this to correlate limb velocities and joint angles with patch-clamp recordings of activity. This will require a major effort which is well beyond the scope of the present Short Report format.

*2) In*
Figure 2*, glutamate spillover currents significantly contributed to the spike outputs of GCs, and subthreshold depolarization strongly correlated with the frequency of burst firing of GCs during movements (*Figure 2*). However in*
Figures 3 and 4*, peak correlation between the motion index and the spillover was fairly low. The authors explain that this is due to slow time course of spillover currents (subsection headed “Linking synaptic input parameters with output spiking during locomotion”). However, the strong correlation between spillover currents and GC spike outputs indicates that the time course of change in CG spike rate is similar to that of spillover currents (*Figure 2*, left). Therefore the correlation between spillover currents and the motion index is expected to be similar to that of GC spikes*.

This is an interesting point, i.e., despite the high correlations (Figure 2) between spillover and fast transmission, the modulation of spillover by locomotion (Figure 4) is lower than for fast transmission. To illustrate what we think is going on, we have made a new supplementary figure (Figure 4—figure supplement 1) in which we show that spillover acts like a smoothing temporal filter. We have taken the bins representing the EPSCs rate in our cells (n = 9), and convolved them with a biexponential kernel with rise time 50 ms and decay time 100 ms (these values were selected to mimic the on-off kinetics of glutamate spillover, see for instance Mitchell and Silver, Nature 2000); please note that the precise values are not important for the qualitative point we are making. Cross-correlating this filtered signal with the original EPSCs signal gives a qualitatively similar cross-correlation as with the actual spillover recorded (compare with Figure 2), with a high peak value of 0.5. However, if we compare the locomotion modulation index of the filtered signal with the original EPSCs, we see that the filtered signal tracks the animal’s step much less well (Figure 4—figure supplement 1), effectively because it is low pass filtering the information at crucial frequencies.

*3) Analyses of the difference between the left and right limbs are interesting. In particular, the direction of modulation seems to be opposite in a single GC (*Figure 4*). It is a pity that the authors only analyzed the absolute depth of modulation and discarded the information about alternation of the left and right limbs. They should analyze the relation between the left-right alternation and GC activities, which should also be related to the laterality of the recording site*.

To address this issue we have added a figure on the lateralization of the step cycle modulation (Figure 4—figure supplement 2). Here we show, by drawing an analogy with orientation tuning in the visual system, that granule cell responses can show modulations at different phases of the step cycle, and that comparing the maximal modulation of one limb with the other shows that these peak modulations tend to happen at around 90 degrees. The direction of modulation (up or down) can vary between cells.

[Editors' note: further revisions were requested prior to acceptance, as described below.]

*The primary issue is whether the data support the conclusion that the granule cell activity is really correlated to “specific attributes” of the motion, as detailed in Reviewer 2's point #2 below. This issue could be most simply addressed by editing the text to curtail the extent of the conclusions*.

We have modified the text to be more conservative in our claims.

Reviewer #1:

*The authors have addressed virtually all issues. One relevant point left, regarding reviewer 1's major point 6: The authors improved their description of the spiking patterns observed, as they now explicitly refer to particularities of burst firing (in the third paragraph of the subsection headed “Whole-cell recordings from granule cells and mossy fibers during locomotion”). Unfortunately, in*
Figure 1*, the authors still present a trace with an instantaneous firing rate of probably well above 100 Hz (*Figure 1*) next to a summary stating a maximum frequency of around 15 Hz (*Figure 1*). I think it would be better to address the burst firing also in the figure itself. All the rest is well done*.

We understand the reviewer’s concern, but we have shown all the recorded bursts in Figure 2, and also described the burst statistics in the text. The purpose of Figure 1 is to show movement-related event rate increases.

Reviewer #2:

*The authors have clarified and addressed the points previously raised. There are some significant findings. Most significant is the recording of granule cell activity in awake animals and the comparison of activity during restful waking and locomotion. The major remaining concern I have is with the interpretation of the results. I think given the low sample sizes for the locomotion recordings (particularly for the mossy fibers, but even for the granule cells) the authors need to be conservative in their conclusions*.

We appreciate the positive feedback and have done our best to tone down the claims that the reviewer finds to be questionable.

*1) In first lines of the Discussion and elsewhere the authors imply that they have “directly measured the transformation from synaptic input patterns to spike output in single neurons”. This statement suggests that they have measured the inputs and outputs to the same cell and that they have defined the transform for individual cells. They have not. What they have done is record two populations of cells and try to infer how activity transforms from one to the other. The problem is that each of these populations is heterogeneous in terms of spontaneous and behavior-related activity. Thus, any number of transforms is possible depending on how MFs connect with granule cells. Even the recordings of granule cell EPSCs and spikes were done for different (though perhaps overlapping) populations of cells, so we cannot say how the EPSCs relate to the specific firing patterns*.

We have changed the text to be more accurate.

*2) The authors state that their results show that the activity of individual granule cells is tuned to “specific features of the animal's motion, rather than merely correlating with just the amount of motion”, and that they can reconstruct the “entire step sequence” (subsection “Linking synaptic input parameters with output spiking during locomotion”). But it seems that their results are more consistent with a general association between changes in granule cell activity and movement*.

*First, the analysis of the video shows good correlations between movement (motion index) and neuronal activity when large time windows (bins of 1.5 s duration) are used but much poorer correlations when “high” resolution windows of 33 ms, which themselves are actually not particularly precise in terms of motor control, are used. Compare*
Figure 3
*but note the different scales of the y axes*.

*Second,*
Figure 3—figure supplement 1
*shows that the increase in neuronal activity begins 100's of milliseconds ahead of the actual movement and decreases to baseline over the course of a second following the cessation of movement. This lack of precise temporal linking of neuronal activity and locomotion is also apparent in*
Figure 1.

*Third, the principle components derived from the video (and used to correlate with neuronal activity) do not seem to be reflecting particularly specific features, at least based on their descriptions. For example, the individual components were said to represent limb, head, and “rest of body” motion. Moreover, the sum of the components representing each of the three categories is 1542, meaning that many of the 1000 components represent more than one category of movement*.

*Fourth, while the analyses shown in*
Figure 4
*certainly show that periods of movements predicted by the neuronal activity generally match up with the actual periods of movement, it does not appear that the individual predicted movements align particularly well with the actual movements. Moreover, for most of neuronal signals only a small minority of cells showed a significant predictive quality, and none showed a predictive quality value approaching 100%, which would have implied a matching of the entire step sequence. Furthermore, the predictive quality measure was only used to look at stance-to-swing transitions. One wonders why the swing-to-stance transitions were not similarly analyzed*.

The reviewer is concerned that we have not sufficiently addressed the possibility that our results are just due to an increase in activity during motion. The strongest argument we have against this is the step cycle-related rate modulations: If the rate merely correlated with the amount of motion, then we wouldn’t see down-modulations of rate during the swing phase of one limb as we observe in some of our cells (Figure 4—figure supplement 2).

The reviewer correctly points out that Figure 3 show that at coarse time scales, the correlation coefficients are high, while the cross-correlation coefficients at fine time scales are smaller. First of all, these two graphs are measuring slightly different things (note there was a mistake in the axis label of Figure 3 which we have corrected): for coarse time scales we are measuring the raw correlations, whereas for the short time scales we are measuring the peak in the normalized mean-subtracted cross-correlation. The mean subtraction implies we have measured modulation in event rate versus modulation in motion index, so by definition these cannot merely be due to average motion. However, we stress that we agree with the reviewer’s main point: there is a general increase in the event rates which is non-specifically related to motion, and there is a smaller modulation of rates which transmits information on top of that. Indeed we would argue that this is the main finding of our paper.

To the reviewer’s third point, it is correct that we have components with several parts of the body represented (this is best illustrated by the examples in Figure 3 which shows regions of activity for the ears and limbs), but what we would argue is that these components often show coherent movement of related body parts (e.g. forelimb motion followed by hindlimb motion representing locomotion at a certain speed). However we agreed with the comments in the previous revision that manual labelling of these components is fraught with subjectivity, which is why we settled on a—less informative—automatic labelling method. We think that the case for relative specificity of the neural activity for a given mode of motion is most strongly emphasized by the comparison with the gray curves generated with surrogate data in Figure 3: this demonstrates that there are clearly a few components which correlate far above chance with neural activity. Since these cross-correlation are again generated by mean subtraction, they cannot be due to a non-specific increase in rates with the general amount of motion. However, we have toned down the references to tuning to address the reviewer’s concerns.

Finally, we picked stance-to-swing transitions since these sequences of movement can most accurately be resolved in time from our videos. Furthermore, our metric (which requires precise timing and good recall/precision) and our algorithm (which is unsupervised) are quite conservative and it would be very surprising to be able to get close to 100% precision with single cell events. Given that granule cells are known to code for many different sensory modalities, it is surprising—or at the very least non-trivial—to be able to read out discrete motor events in the way we have shown. It is quite likely that if we optimized a supervised learning method to read out the step sequence, we would be able to improve the prediction score, but we feel this is beside the point we are trying to make, namely that the step sequence is represented in the granule cell layer, which is an entirely novel finding. Probably the most principled way to estimate just how much of this information is encoded would be to calculate the mutual information or transfer entropy between the step sequence and the neural activity, but these methods would require very long recording times to converge. Nevertheless we have modified the text to be more conservative in our claims about step prediction in a way we hope will satisfy the reviewer and editor.